



# A New Database of Extreme European Winter Windstorms

Clare Marie Flynn[1, 2], Julia Moemken[3], Joaquim G. Pinto[3], and Gabriele Messori[1,2,4]

[1]Department of Earth Sciences, Uppsala University, Uppsala, Sweden
[2]Swedish Centre for Impacts of Climate Extremes (climes), Uppsala University, Uppsala, Sweden
[3]Institute of Meteorology and Climate Research Troposphere Research, Karlsruhe Institute of Technology (KIT), Karlsruhe, Germany
[4]Department of Meteorology and Bolin Centre for Climate Research, Stockholm University, Stockholm, Sweden

**Correspondence:** Clare Marie Flynn (clare.flynn@geo.uu.se)

**Abstract.** European windstorms pose a significant threat to people, infrastructure and the natural environment. Several windstorms in the recent past have caused substantial damages, and losses associated with extreme windstorms may increase with climate change. Characterizing the footprints of destructive windstorms is thus key to providing quantitative estimates of storm-related economic losses. To that end, we have developed a new, publicly available database of extreme European wind-
storm footprints for the extended winter season during 1995-2015. In contrast to previously compiled European windstorm databases, we include storm footprints derived from four different data sets, rather than a single source: the ERA5 reanalysis, the COSMO-REA6 reanalysis for Europe, the COSMO-Climate Limited-area Mode regional climate model driven by ERA5 on the EURO-CORDEX domain, and simulation output from the same model but on an enlarged Germany domain with higher horizontal resolution. The database includes both the footprints themselves, expressed as the relative daily maximum wind
gusts associated with a storm event, and the daily maximum wind gusts in absolute magnitude associated with the footprints. We derived and included the storm footprints associated with the 50 most extreme storms, or Top50 storms, identified within each of the four input data sets. We applied a consistent methodology, the storm loss index, across input data sets for identifying storm footprints and assessing their severity. This enables a direct comparison between the footprints derived from the different input sources, eases future efforts to extend the time record of the database or to include additional input data sets,
and enables assessment of uncertainty in the footprints. Moreover, since we derived the Top50 storms from each input on its native horizontal resolution, the database also allows to characterize the impact that horizontal resolution can have on footprint identification and severity assessment. Our database thus supports both the research community and the insurance industry in exploring the data set and resolution dependence of assessments of extreme storm hazards.

## 1 Introduction

Winter windstorms constitute the most costly natural hazards for Europe, posing a significant threat to people, infrastructure and the natural environment (Mitchell-Wallace et al., 2017; Priestley et al., 2018; Pinto et al., 2019; Walz and Leckebusch, 2019; Munich Re, 2022; Moemken et al., 2024). Average annual windstorm insured losses in Europe are on the order of several billion USD, with total losses estimated to be well in excess of this. Windstorms also lead to important non-monetary





losses such as casualties (Schwierz et al., 2010; Priestley et al., 2018; Gliksman et al., 2023). Windstorm damage scales non-
linearly with storm intensity, and single storms of unusual severity can cause losses which exceed the long-term annual average.
For example windstorm KYRILL, the most severe storm of the 2006/2007 winter season and one of the most severe within
the past three decades, struck Europe in January 2007 (Fink et al., 2009; Priestley et al., 2018) and alone caused 5.8 billion
USD in insured losses. It further led to 54 fatalities, significant disruptions in transportation, and electricity outages, among
other impacts (Deutsche Rück, 2008; Munich Re, 2015). KYRILL was also the strongest storm within a series of intense
windstorms that struck Europe during the 2006/2007 season, which caused an estimated 10 billion USD in cumulative insured
losses for this season as well as additional injuries and deaths (Pinto et al., 2014). Indeed, serial clustering of European winter
windstorms, namely when multiple storms follow a similar track in quick succession (Dacre and Pinto, 2020), can magnify
losses relative to individual storm events. Moreover, more severe storms are more likely to cluster than less severe storms (Pinto
et al., 2014; Priestley et al., 2018). Though uncertainties remain in predictions of changes in windstorms with climate change,
climate model evidence suggests an increase in both the frequency and strength of storms over northern and central Europe,
particularly for more extreme windstorms, with a corresponding increase in storm losses (Pinto et al., 2007; Leckebusch et al.,
2007; Schwierz et al., 2010; Pinto et al., 2012; Little et al., 2023; Severino et al., 2024). This occurs in spite of the well known
general decrease in the total number of extratropical cyclones over the region (Ulbrich et al., 2009; Priestley and Catto, 2022).
Characterizing and understanding extreme European winter windstorm losses is therefore a highly socioeconomically relevant
goal.

To address this need, several publicly available and subscription-based storm loss databases have been created and main-
tained. The publicly available loss databases are often based on a storm severity index that incorporates meteorological in-
dicators and empirical approximations for insured losses. A widely used index is the storm loss model based on Klawa and
Ulbrich (2003), in which the predicted windstorm damage is proportional to the cube of the exceedance of the wind speed
over a relative threshold value and to the population density (Klawa and Ulbrich, 2003; Pinto et al., 2012; Gliksman et al.,
2023; Moemken et al., 2024). Other approaches are also employed, such as statistical downscaling of wind data (van den
Brink, 2019). Insurance and reinsurance companies also simulate storm losses with catastrophe models. Index-based loss es-
timates and industry-computed insured losses based on catastrophe modelling, however, both rely on accurate wind speed or
gust data with sufficient spatial and temporal coverage. Windstorm damage is typically assumed to arise from the strongest
wind speeds or gusts encountered during the storm, such that estimates of storm-related losses rely on identifying the loca-
tions impacted by a given storm as well as the peak winds as accurately as possible (Klawa and Ulbrich, 2003; Leckebusch
et al., 2007; Pinto et al., 2012, 2014; Priestley et al., 2018; Cusack, 2023; Gliksman et al., 2023). The occurrence exceedance
probability (OEP) and the annual exceedance probability (AEP), two metrics based on loss estimates that are commonly used
in the insurance industry, further rely on accurate wind data. The OEP and AEP represent the maximum loss event during
a season and the total losses summed over all events during a season, respectively, and are used to assess return periods of
extreme storms and the impact of storm clustering on losses (Priestley et al., 2018). The only loss data that are not sensitive to
the quality of the meteorological data are industry reports of recorded insured losses, such as those collected in the PERILS
database (http://www.perils.org). PERILS is based on data gathered by insurance and reinsurance companies, but only provides





information on an annual subscription basis and for a subset of European countries (Gliksman et al., 2023; Moemken et al., 2024).

Wind data of a sufficient quality are therefore crucial to producing reliable, openly accessible and comprehensive storm loss estimates and for the assessment of future risks. This wind information is often provided as a storm footprint, essentially a map of the peak winds and their magnitudes encountered during a storm event at locations affected by the storm. These are typically determined through use of a wind exceedance threshold such as that used in storm severity indices (Klawa and

Ulbrich, 2003; Pinto et al., 2012, 2014; Priestley et al., 2018). Two publicly-available databases which provide storm footprints and the corresponding storm loss estimates at the time of writing are the eXtreme Wind Storms Catalogue (XWS; Roberts et al. (2014)) and the Copernicus Climate Change Service (C3S; C3S Climate Data Store (2022); van den Brink (2019)) databases. These contain the most extreme European winter windstorms over the past approximately 40 years and have proven useful in the study of extreme windstorm impacts; more detail on these databases is given in Section 2. However, the XWS and C3S

databases are each derived from a single input data set; the ERA-Interim (Dee et al., 2011) reanalysis in the case of XWS and ERA5 (Hersbach et al., 2020) for C3S. While ERA5 is the current state-of-the-art reanalysis data set, there are indications that storm footprints based on ERA5 peak near-surface wind gusts may contain inaccuracies (Cusack, 2023), while ERA-Interim is known to have deficiencies in representing many storms during the later 20th and early 21st centuries (Moemken et al., 2024). Furthermore, a downscaling approach involving the use of an atmospheric or statistical model was employed to derive the

footprints and estimated losses for both databases, requiring intensive computational resources for database extension. Finally, different definitions were used in each database for the spatial extent of the footprint (Roberts et al., 2014; van den Brink, 2019), making the two not comparable, and thus hindering assessment of the uncertainty associated with footprint computation. Thus, while XWS and C3S constitute invaluable tools in the study of extreme windstorms, a need remains for a storm footprint database based on multiple meteorological data sets which are processed with a standardised methodology.

One of the identified challenges in terms of windstorm research is to have a consistent, reliable and extendable database for historical windstorm risk (Pinto et al., 2019). In this manuscript, we present a new database of the 50 most extreme European winter windstorms derived from four different input data sets with four different native horizontal resolutions using a standardised methodology. The meteorological data comes from two reanalysis data sets, ERA5 (Hersbach et al., 2020) and COSMO-REA6 (Bollmeyer et al., 2015), and output from two regional climate model simulations, CCLM_ERA5_EUR-11 and

CCLM_ERA5_CEU-3, as further described in Section 2. This database therefore complements the existing storm databases and expands the footprint data available for use by the scientific community and industry. Its design facilitates an assessment of the impact of different data sources and different horizontal resolution on identification of extreme storms and their estimated losses. Equally importantly, an estimate of the uncertainty associated with the footprint itself could also be assessed given the diversity of input data sets used in our database, as similarly enabled by a new database for storm loss estimates from multiple

perspectives (Moemken et al., 2024). This database may therefore be used by the scientific community and insurance industry to address the crucial question of what "quality" of wind gust data is required for accurate and reliable characterization of extreme windstorms and their impacts, supporting the need for better analysis of storm damage to reduce their impacts on society.



Throughout the rest of this study, we use the terms "windstorm" and "storm" interchangeably to refer to the extreme storms
identified in each data source that affected Europe during the extended winter season. Storm names, when they are given names
rather than dates, are capitalised.

## 2 Methods

### 2.1 Overview of the database

Our European windstorm database was derived from four different input data sets with different horizontal resolutions. It
consists of the windstorm footprints as represented by the relative daily maximum wind gusts during the storm event (unitless),
their associated daily maximum wind gusts in absolute units ($ms^{-1}$), the loss index ($LI_{3D}$; unitless) integrated over a Core
Europe region (42 °N–60 °N, 10 °W–15 °E), and the unintegrated loss index at each grid point (unitless). The unintegrated loss
index enables users to compute an integrated loss over a subset region of a given storm's footprint, such as at the country level.
We further provide the name and dates of occurrence of each windstorm, their ordinal rank, and their relative rank (also termed
the normalized loss). Because the input data sets are on different native horizontal grid types and resolutions, we maintained
the native grid information and created a separate database netCDF file for each input source rather than merging all into one
file.

The database covers extended winter seasons (October to March, ONDJFM) from January 1995 to December 2015, a 20-
year period covered by all input data sets, and is based on daily wind gust maxima derived from hourly wind gust data. The
fifty most extreme storms, or Top50 storms, were identified for each input data set based on the Core Europe integrated loss
index and included in the database. The individual storms that make up the Top50 storms for each input source therefore differ,
and in some cases the same storm may have somewhat different dates of occurrence across the input data sets.

### 2.2 Input Data Sets

Table 1 summarizes the input data sets used to create the new database presented here. These consist of two reanalysis data
sets, ERA5 (Hersbach et al., 2020) and COSMO-REA6 (Bollmeyer et al., 2015), and output from two regional climate model
simulations on different domains, the CCLM_ERA5_EUR-11 and CCLM_ERA5_CEU-3 simulations, performed with the
COSMO model in climate mode (Rockel et al., 2008). These data sets differ in horizontal resolution, spatial domain sizes and
types of horizontal grids, such as a regular latitude-longitude or curvilinear grid.

The publicly available ERA5 reanalysis data set from the European Centre for Medium-Range Weather Forecasts (ECMWF)
provides global data with a 0.25° horizontal resolution and with many parameters available on an hourly temporal resolution
(Hersbach et al., 2020). We analyse ERA5 storm footprints over 27-72° N and 22° W-45° E. The reanalysis is based on
the Integrated Forecasting System (IFS) Cy41r2 model (Hersbach et al., 2020), which was operational at ECMWF in 2016
(Bonavita et al., 2016). Data span 1940 to near-present. ERA5 is often used for evaluation and bench-marking of other data



sets, such as model simulation output, due to its global coverage, high temporal resolution, and observationally constrained, physically consistent atmospheric data.

The Hans-Ertel-Centre for Weather Research (HErZ) and the Deutscher Wetterdienst (DWD; German Weather Service) developed the COSMO-REA6 (Consortium for Small-scale Modelling-Reanalysis, 6 km) high-resolution regional reanalysis data set for Europe, with a 0.055° (approximately 6 km) horizontal resolution and 15-minute or hourly temporal resolutions (Bollmeyer et al., 2015). COSMO-REA6 is available over 1995-August 2019, since it uses ERA-Interim reanalysis data (Dee et al., 2011) for lateral boundary conditions and the latter is no longer produced. The spatial domain matches that of the CORDEX EUR-11 domain (Jacob et al., 2014, 2020), which covers approximately 27-72° N and 22° W-45° E. Unlike ERA5, COSMO-REA6 is based on the COSMO (Consortium for Small-scale Modelling) model in numerical weather prediction mode, developed by the DWD (Baldauf et al., 2011). It uses nudging of surface synoptic conditions, aircraft measurements, radiosondes, buoys, ship reports, and wind profilers as its data assimilation scheme (Bollmeyer et al., 2015).

The CCLM_ERA5_CEU-3 simulations were run with the non-hydrostatic COSMO model in climate mode (COSMO-Climate Limited-area Mode or COSMO-CLM) regional climate model, version 5.0 (Rockel et al., 2008), by the DWD in collaboration with the Climate Limited-area Community (CLM-Community) collaborative network. COSMO-CLM is the climate version of the limited-area numerical weather prediction COSMO model used to produce COSMO-REA6, convection-permitting, and driven by ERA5 through direct downscaling. CCLM_ERA5_CEU-3 spans an enlarged Germany or COSMO-DE domain (approximately 45-58° N and 1-20° E) at 0.0275° (approximately 2.8 km) horizontal resolution (Baldauf et al., 2011; Brienen et al., 2022). Output is available for the years 1979-2019 up to hourly resolution (https://esgf.dwd.de/search/esgf-dwd/). Output does not extend beyond 2019 as the DWD discontinued use of the COSMO-CLM model. The CCLM_ERA5_CEU-3 simulation output used to create the extreme storms database presented here therefore represents the highest horizontal resolution, but smallest spatial domain, input data source of the four input data sets we used.

Lastly, the CCLM_ERA5_EUR-11 simulations were performed by the Helmholtz Center Hereon in collaboration with the CLM-Community and EURO-CORDEX. The COSMO-CLM model in the same configuration as for the CCLM_ERA5_CEU-3 simulations was used, but on the CORDEX EUR-11 domain at a horizontal resolution of 0.11° (approximately 12 km). The European branch (EURO-CORDEX; Jacob et al. (2014, 2020)) of the Coordinated Regional Downscaling Experiment (CORDEX; Giorgi and Gutowski (2015)), is a collaborative initiative that seeks to advance regional climate and Earth system science in Europe. It defined the CORDEX EUR-11 domain in 2013, intended to be used for simulations with a horizontal resolution of 0.11°. Output is available for the years 1979-2020 at up to hourly resolution.

## 2.3 Windstorm Footprint Identification

To identify the windstorm footprints within each input data set and to quantify their severity, we employed the storm loss model known as the loss index (LI). This was developed by Pinto et al. (2012) and extended by Karremann et al. (2014), and is in turn based on the storm loss models developed by Klawa and Ulbrich (2003) and Leckebusch et al. (2007). The LI was originally developed based on near-surface daily maximum wind speed (Pinto et al., 2007), but we exchanged this for near-surface daily





maximum wind gusts derived from hourly data. This is an unproblematic exchange, as wind speeds are often used as a proxy for wind gusts when gust data were unavailable, and maintains consistency across calculations and comparisons.

The LI is based on the assumption that storm damage occurs only for the highest 2% of local wind speeds, or wind gusts in our case (Klawa and Ulbrich, 2003; Pinto et al., 2012; Karremann et al., 2014). The footprint of an individual storm is therefore only those locations whose local daily maximum wind gust on the date of the storm exceeds the local 98th percentile of wind gusts computed over the full 20-year record. The storm footprint is expressed in our database in terms of the unitless relative wind gust; the absolute wind gusts associated with the footprint are also given in our database. The relative wind gust is the ratio of the local daily maximum wind gust that occurred during a given storm to the local 98th percentile, which indicates the

magnitude of the exceedance over the 98th percentile and is everywhere greater than 1.0 within the storm footprint (Pinto et al., 2007, 2012).

    Individual storms must then be separated from each other in time and their severity assessed. This is accomplished through calculation of the LI itself, which relies on two additional assumptions. First, potential storm losses are assumed to increase with the cube of the maximum wind speed or gust, as this is proportional to wind kinetic energy. Second, storm losses are

170 linked to the exposure/insured value, which can be approximated by population density (Pinto et al., 2012; Karremann et al., 2014). We used the Gridded Population of the World, version 4 (GPWv4) population density data set for the year 2020, the latest year available, provided by the Center for International Earth Science Information Network (CIESIN) at Columbia University (Center for International Earth Science Information Network - CIESIN - Columbia University, 2018). These data have a horizontal resolution of $0.04°$, and were regridded to the resolutions of the four input data sets before use. The LI was

175 computed for each day, just as the daily maximum wind gust, and is defined as follows:

$$LI = \Sigma_{ij}[(\frac{v_{ij}}{v_{98_{ij}}})^3 * POP_{ij} * I(v_{ij}, v_{98_{ij}})] \tag{1}$$

where

$$I(v_{ij}, v_{98_{ij}}) = \begin{cases} 0, & \text{for } v_{ij} \leq v_{98_{ij}} \\ 1, & \text{for } v_{ij} > v_{98_{ij}} \end{cases}$$

and indicates whether or not the daily maximum wind gust at grid point $ij$ falls within the storm footprint. $POP_{ij}$ is the

180 population density at grid point $ij$, and $v_{ij}$ and $v_{98_{ij}}$ are the daily maximum wind gust and the 98th percentile wind gust at grid point $ij$, respectively (Pinto et al., 2012; Karremann et al., 2014). An overlapping three-day sliding time window is applied to the LI in order to separate individual storms in time, and the temporal local maximum of each three-day window is assumed to be the individual storm event (Karremann et al., 2014):

$$LI_{3D} = \Sigma_{ij}[([max_{3D}\frac{v_{ij}}{v_{98_{ij}}})]^3 * POP_{ij} * I(v_{ij}, v_{98_{ij}})] \tag{2}$$

The $LI_{3D}$ provides the final, temporally separated storm footprints for individual storms. It should be noted, however, that if storms occur too closely together in time, such as storms LOTHAR and MARTIN in December 1999, it can be very difficult or impossible to separate the storms with the $LI_{3D}$; this seldom happened in the creation of our database, but when it did occur,



the storms were typically counted as one "combined" storm in the database. The $LI_{3D}$ can be integrated over any spatial domain of interest; to create our database, we integrated over the Core Europe region defined above, following Pinto et al. (2012). It should be further noted that the unintegrated loss index included in the database refers to the summand inside the $\Sigma$ operator in Equation 2 that is computed at each grid point before any integration is performed, and not to the $LI_{3D}$. This quantity is thus akin to a pre-$LI_{3D}$.

An example of a storm footprint as represented by the relative wind gusts is shown in Figure 1 for storm KYRILL for each of the four input data sets, as KYRILL was identified as the most extreme storm in all four input sources. Figure 2 displays the absolute daily maximum wind gusts associated with the footprints for each of the four input sources as well as for XWS and C3S. Figures 1 and 2 clearly highlight the differences among the footprints for the same storm due to differences in input data sets, and, in the case of Figure 2, storm footprint definition.

## 2.4 Selection of Extreme Windstorms

We selected the 50 most extreme (Top50) storms over the 1995-2015 period derived from each input data set based on the magnitude of the $LI_{3D}$ integrated over Core Europe. The magnitudes were ranked from largest to smallest, and the fifty largest magnitudes were chosen. The selected storms were manually checked to ensure that they represented either unique individual storms, or, in the case of storms that could not be sufficiently separated, a unique "combined" storm. The three-day time windows containing the fifty largest $LI_{3D}$ magnitudes were associated with unique dates per dataset, and the dates of the mid-point of each window became the storm dates of occurrence for that dataset in the database.

Based on these dates, storm names were assigned to the individual storm events. Storm names were taken from several sources, including the lists of named storms produced by the DWD and the Freie Universität Berlin for the years 1999-2015 (https://www.wetterpate.de/namenslisten/tiefdruckgebiete/index.html; in German), the past European winter windstorm documentation provided by Deutsche Rück for the years 1997-2015 (https://www.deutscherueck.de/downloads, in German), and from Wikipedia articles about European winter windstorms (primarily English articles). Some storms, primarily those occurring earlier than 1997, appear to lack names given by a meteorological service or research institution; in these cases, we have taken the storm date of occurrence as the storm name preceded by the lower case letter "u," and we have preferred the date as identified from ERA5 for the name, if the storm was identified within ERA5 and at least one other data set. As mentioned above, some storms could not be sufficiently separated with $LI_{3D}$ in some input data sets, and were thus taken as a single, "combined" storm and the names of the individual storms that could not be well separated were hyphenated to create the storm name. Some individual storms had two different names given by different entities, so these storms retained both names, one "main" name and a "secondary" name given in parentheses (or connected by an underscore rather than parentheses within the database files for ease of coding). Lastly, two different storms were given the same name, "Franz," and so we have identified one storm as FRANZ (2007-01-11) and the other as FRANZ-II (1999-12-12); the hyphenated name in this one case does not indicate two storms that could not be sufficiently separated.



## 2.5 Existing Windstorm Databases for Comparison

The XWS (Roberts et al., 2014) and C3S (C3S Climate Data Store, 2022; van den Brink, 2019) storm databases were used as comparison data sets for the new database we describe here; these databases are also summarized in Table 1.

The publicly-available XWS database (http://www.europeanwindstorms.org) includes 50 of the most extreme European storms within the extended winter season for the years 1979-2014, and includes storm footprints and storm severity indices or loss estimates; this database will not be extended beyond 2014. The footprints were computed through dynamically downscaling the previous generation ERA-Interim reanalysis data set (Dee et al., 2011) to a horizontal resolution of 0.22° (approximately 25 km) with the UK Met Office Unified Model (MetUM; Davies et al. (2005)) over a domain including western Europe and the eastern North Atlantic. The footprints are defined as the maximum 3-second wind gusts at each grid point in the downscaled domain over a 72-hour period, but, rather than taking the maximum gusts over the entire domain, all wind gusts outside of a 1000 km radius centered on the storm track are neglected before taking the maximum. This is done to separate storms that occurred closely together in time from each other. A meteorological severity index $S_{ft}$ (Roberts et al., 2014) was computed for each storm as:

$$S_{ft} = (U_{max})^3 * N \tag{3}$$

where $(U_{max})^3$ is the cube of the storm's maximum near-surface wind speed as an indication of storm intensity and $N$ is the footprint size index. Though differently formulated, the $S_{ft}$ index is based upon the winds associated with a storm, just as the LI and $LI_{3D}$. Of the 50 storms in the XWS database, 23 storms were included after consultation with Willis Research Network and based on the extreme values of insured losses and constitute the "insurance storms," while the remaining 27 storms were included based on the $S_{ft}$ index. The insurance storms are provided with an insured loss amount in USD (indexed to the year 2012) in addition to the $S_{ft}$ index. For comparison to our database, we ranked the 50 XWS storms based on their $S_{ft}$ magnitudes. The XWS data were used directly and required no further processing, as this database has already restricted their footprints to the area affected by each storm.

The publicly-available C3S database (https://cds.climate.copernicus.eu/) was derived from ERA5 and includes significant winter storms from the years 1979-2021 for 21 countries in western, central, and northern Europe on a horizontal resolution of 1.0 km; southeastern European countries were not included. Though this database currently extends to 2021, it may be extended further as the ERA5 reanalysis will continue to be updated. Similarly to XWS, the C3S footprint was defined as the maximum 3-second near-surface wind gust over the 72-hour period capturing the storm, but the footprints were computed through statistically downscaling ERA5 data instead. A multiple linear regression model following van den Brink (2019) was developed and validated, and is based on the ERA5 wind gust data, wind gusts estimated from the wind speed (also taken from ERA5) shear between the 10 m and 100 m altitude levels, and station elevation height to derive estimates of the strongest wind gusts during a storm period. This method is valid only for land areas and allowed the C3S database to estimate the strongest wind gusts for locations far from or in between weather observation stations. A total of 148 storm footprints are included in this database. However, the strongest wind gusts associated with a storm were estimated over all land areas within the full C3S domain, covering 35-70° N and 20° W-35° E. Though the footprint is often apparent as the area crossing Europe with





the largest wind gusts, this leaves ambiguity over precisely where the footprint begins and ends. Because the C3S database
is derived from ERA5, we used the footprints we identified from the ERA5 data set as a spatial mask to "cut out" the C3S
wind gusts belonging to a given storm's footprint, after first regridding the ERA5 footprints to the C3S horizontal grid. We
then computed the $LI_{3D}$ for each C3S storm footprint using the regridded ERA5 footprints to assess storm severity, in order to
maintain consistency between the footprint area and severity index.

We selected only those storms from the XWS and C3S databases from the 1995-2015 period that were also found in at least
one of the four input data sets listed above, as these databases were used for comparison only, leading to 23 extreme windstorms
for XWS and 30 for C3S. It should be noted that these databases did not provide relative wind gust data, and therefore direct
comparisons were possible between the XWS and C3S databases and our database only for the absolute wind gusts associated
with the footprints.

### 2.6 Ordinal and Relative Ranking of Windstorms

In order to compare storm severity among the storms identified within our database, the Top50 storms were each assigned
an ordinal rank and a relative rank, also called the normalized loss index or simply normalized loss. The ordinal ranking is
straightforwardly based on the magnitude of the $LI_{3D}$, where the storm with the largest $LI_{3D}$ was assigned rank 1, and the storm
with the smallest $LI_{3D}$ was assigned rank 50. Ordinal ranks for the same storm identified in two or more input data sets can
differ.

The relative ranks, or normalized losses as it will be referred to in the following sections, is based on min-max scaling of the
$LI_{3D}$:

$$normalized loss_i = \frac{LI_{3D_i} - min(LI_{3D})}{max(LI_{3D}) - min(LI_{3D})} \tag{4}$$

where $i$ refers to the value for an individual storm. The normalized loss ranges from 0.0 for the storm with the smallest impact
(the storm with ordinal rank 50) to 1.0 for the storm with the most impact (the storm with ordinal rank 1); the normalized
losses and the ordinal ranks vary inversely and monotonically with each other. The normalized loss thus expresses the severity
of each individual storm as relative to the severity of the most extreme storm (that with both ordinal and relative rank of 1)
and provides an indication of how different the storms are from each other. These ordinal and relative ranks are included in
the netCDF files that constitute our extreme storms database. The ordinal ranks and normalized losses were also computed for
the XWS and C3S storms we use for comparison below, though relative to a total storm number of 23 for XWS and 30 for
C3S rather than 50. We also based the ranks and normalized losses on the $S_{ft}$ index for XWS, and on the $LI_{3D}$ magnitudes
corresponding to the ERA5 footprints used to "cut out" the footprints for C3S.



## 3 Results

### 3.1 Overview of the Identified Extreme Storms

Our database identified 76 unique storms within the Top50 storms across the four input data sets. Of these storms, 29 storms
(approximately 38% of the unique storms) were identified within all four input sources, constituting the common storms listed
in Table 2. Table 2 also indicates whether these storms are found in the XWS and C3S databases. The supplemental file
Top50Storms_All_Summary.csv summarizes all the Top50 storms rather than only the common storms identified within each
of the input data sets and whether they are found in XWS and C3S.

Despite occurring closely together in time, only storm LOTHAR belongs to the common storms while MARTIN does not,
as it was not identified within the COSMO-REA6 and CCLM_ERA5_CEU-3 data sets; this is likely because MARTIN could
not be sufficiently separated from the first-occurring LOTHAR within the COSMO-REA6 data set, while MARTIN's more
southerly storm track fell mostly outside the domain of CCLM_ERA5_CEU-3. The date of occurrence for MIKE-NIKLAS
was identified as 2015-04-01 in CCLM_ERA5_EUR-11. Though technically outside the temporal domain we considered for
the database, we decided to keep this storm for this input source as MIKE-NIKLAS was identified within the remaining input
sources as occurring within the extended winter season we defined.

This leaves 47 storms (approximately 62% of the unique storms) that were identified in at least one of the input data sets but
not all four (listed in the supplemental file Top50Storms_All_Summary.csv). Of these 47 storms, 15 were identified in only
one input source, whereas the remaining 28 storms were identified in two or three input sources. Four storms were identified in
ERA5 alone (BECKY, the BOXING DAY STORM, FRIEDHELM, and JETTE); six were identified in CCLM_ERA5_EUR-
11 alone (DAGMAR, EBERHARD, JULIA, u19961106, u19961120, and u20000209) and in CCLM_ERA5_CEU-3 alone
(DORIAN, ELIZABETH, EX-HURRICANE GONZALO, INGO, and QUINTEN); and two were identified in COSMO-REA6
alone (SUSANN and ORKUN). It is unexpected that CCLM_ERA5_CEU-3 rather than COSMO-REA6 displayed a similar
number of storms identified only in that input data set as ERA5 and CCLM_ERA5_EUR-11, given its much smaller spatial
domain. The remaining storms within our database were identified in two or three of the input data sets, and no systematic
pattern appears to exist in which combinations of input sources are preferred.

All common storms, with the exception of four storms, were identified as occurring on the same date across the four input
data sets. The remaining four common storms displayed a discrepancy in date of occurrence of one day (Table 2). When
considering all Top50 storms identified in at least two input data sets, approximately 74% of storms displayed no discrepancy
in date of occurrence and approximately 26% of storms displayed a discrepancy of one day. One of the reasons for these
discrepancies is the smaller spatial domain of CCLM_ERA5_CEU-3 compared to the others.

Only 12 of the common storms were also found in the suite of XWS storms examined here (approximately 52% of all the
XWS storms used here), and 16 of these storms (or approximately 53% of all C3S storms used here) for C3S. When a common
storm was found within both the XWS and C3S databases, the dates of occurrence between XWS and C3S agreed with each
other for all storms except EMMA and XYNTHIA, for which they displayed a difference of one day (Table 2). Relative to
our database, XWS and C3S displayed most often displayed no discrepancy in the date of occurrence for both the common





storms and for all Top50 storms. The differences in dates of occurrence display no preference for before or after the dates in our database.

The storms that are not common to all four data inputs in our database thus demonstrate disagreement on which storms are identified as belonging to the Top50, and mild disagreements in the dates the storms occurred for those occurring in two or three

data sets. Even among the common storms, mild variations in dates of occurrence were not eliminated. These discrepancies exist despite the use of a consistent methodology. This points to the important differences resulting from use of different input data sets, while comparisons with XWS and C3S further highlight potential impacts resulting from differences in storm footprint identification methodology.

## 3.2 Storms per Extended Winter Season

The distribution of storms per extended winter season over all Top50 storms identified from each of the four input data sets, XWS and C3S is displayed in Figure 3. The winters during the second ten years of our database exhibit less storm activity relative to the first ten years; approximately 60% of all Top50 storms occurred during the first ten year period within each of the four input data sets, compared to approximately 40% during the second ten year period. However, important differences among input data sets in how the storms are distributed across winters also appear. For instance, CCLM_ERA5_EUR-11 and

CCLM_ERA5_CEU-3 display a greater percentage of storms occurring during the first half of the time record (60% and 64%, respectively) than do ERA5 and COSMO-REA6 (58% and 54%, respectively).

Further, the input data sets demonstrate disagreement on the periods of low and high storm activity (Table 3). The input sources agree best on the winters with no extreme storm activity: no storms were identified for the 1995/1996 and 2012/2013 extended winter seasons from any of the four inputs, while one storm was identified within CCLM_ERA5_CEU-3 for the

335 2005/2006 (DORIAN) and 2008/2009 (QUINTEN) winters but not within the remaining three inputs. This level of agreement cannot be found for other periods of low or high storm activity. There are no additional winters for which all four input sources agree as belonging to periods of low storm activity, high storm activity, and very high storm activity, as indicated by winters during which 2%, 8%, and 10+% of all storms occurred, respectively (Table 3). While the data sets agree for one winter within each of these periods, the remaining winters show agreement only among two input data sets or are unique to that data set. For

example, the ERA5, CCLM_ERA5_EUR-11, and CCLM_ERA5_CEU-3 data sets agree that the 1999/2000 winter exhibits very high extreme storm activity. However, 10% of all storms occurred during this winter in ERA5 and CCLM_ERA5_CEU-3, while 12% of all storms was obtained for CCLM_ERA5_EUR-11, demonstrating disagreement on the number storms during this winter (Figure 3; Table 3). It is notable that the two winters with very high activity displayed by COSMO-REA6 occur during the second ten years of our database, whereas the very high activity winters for the remaining three input data sets occur

during the first ten years.

The XWS and C3S databases display quite different distributions of storm frequencies compared to our database (Figure 3; Table 3). Though these two databases contained no extreme storms during the 1995/1996, 2005/2006, and 2012/2013 winters, in agreement with our database, they each contain four additional winters during which no extreme storms took place. It is notable that XWS and C3S agree on three of these additional winters (Table 3). XWS and C3S further display no winters that



belong to the low storm activity category, and C3S displays no winters that belong to the high activity category. XWS displays four high activity winters, agreeing with our database for two of these winters. Neither XWS nor C3S exhibit better agreement with our database, given the similar mean absolute differences in storm percentage per winter between XWS or C3S and our input data sets, and the large standard deviations (Table 4). The differences with our database partially reflect that we neglected storms identified within XWS and C3S that were not identified within our database, and that XWS and C3S did not contain all

the extreme storms identified within our database. The differences in storm frequency distribution and periods of high and low storm activity among our database, XWS, and C3S further highlight the influence of horizontal resolution, domain size, and methodology on extreme storm identification and characterization.

### 3.3  Variations in Relative Extreme Storm Severity

#### 3.3.1  Relative Severity per Winter Season

The interannual variability in winter severity within each data set is shown in Figure 4, where the proportion of total losses occurring during each winter is indicated. In agreement with the storm frequency distributions discussed above, a larger proportion of total storm losses within each input data set took place during the first ten years of our database, with similar proportions of total losses during the first and second ten years as for storm frequencies. CCLM_ERA5_EUR-11 and CCLM_ERA5_CEU-3 again display a greater proportion of total losses occurring during the first ten years than do ERA5 and COSMO-REA6. The

proportion of total losses per winter within an input data set generally increases with the proportion of storms per winter, as exhibited by the increases in the proportions of total losses with storm activity per winter for each input source in Table 3. All four input data sets display a statistically significant Pearson's correlation coefficient ($p < .001$) between 0.95 and 0.98 for the correlation between the percentage of all storms per winter and the percentage of total losses per winter. These statistics imply that each input data set contains few winters during which the losses were caused by a single, exceptionally strong storm rather

than by several, weaker storms, as this would weaken the correlation.

This is further supported by the OEP/AEP ratio for each winter season and input data set displayed in Figure 5. The OEP/AEP ratio indicates the degree to which a single storm event dominated that winter season's aggregated loss (Priestley et al., 2018). A ratio much smaller than 1.0 indicates that multiple storm events contributed to the total seasonal losses, while a large ratio closer to 1.0 indicates that a single storm contributed most of the losses incurred during that season; a ratio exactly equal to

1.0 is achieved during winters containing only one storm. If we disregard the winters with a single storm, very few winters display a ratio greater than approximately 0.7 for each input source. This indicates that, for most winters that contained more than one extreme storm, the losses comprise contributions from multiple storms. Though the losses from a single storm during such winters sometimes contribute approximately 50-60% to the aggregated seasonal losses, it is unusual that an exceptionally severe storm contributes an overwhelming share of losses within our database (except for years with a single storm).

However, some variability does exist in average storm severity during a given winter. For example, during the 2006/2007 season, fewer storms caused a proportion of damages similar to the higher storm activity 1996/1997 and 2004/2005 winters in CCLM_ERA5_CEU-3, and similar damage proportions to the higher storm activity 2007/2008 winter in ERA5 and



CCLM_ERA5_EUR-11 (Table 3). This implies that the storms identified within CCLM_ERA5_CEU-3 for the 2006/2007 season are more severe relative to the other input data sets for this winter, as well as to many other winters within CCLM_ERA5_CEU-
3. As with the storm frequencies per winter, the disagreements among input data sets for the severity of any given winter are also apparent in Figure 4. As the same methodology was employed, these could be due, at least in part, to differences in horizontal resolution and spatial domain impacting storm identification and severity assessment.

Again, neither XWS nor C3S presented a better average comparison with our database for the proportions of total losses per winter, given the large standard deviations (Table 4). The storms within XWS for the 1997/1998 and 1998/1999 winters are
likely weaker than the storms within the high activity category winters within our database given the smaller proportions of total XWS losses (Table 3). However, it is notable that the loss proportions for the XWS and C3S winters within the very high activity winter category are generally similar to those from our database, as are the remaining two high activity winters within XWS. Both XWS and C3S further exhibit two winters of unremarkable storm activity, belonging to neither the low nor high activity categories, but large damages, implying stronger storms compared to our database for those winters. Variations in storm
severity between XWS and C3S and relative to our database likely reflect differences in storm identification methodology that impact the storm severity assessments.

### 3.3.2   Variations in Relative Common Storm Severity

Differences in relative storm severity among input data sets are further highlighted in Figure 6 for the common storms only, and including XWS and C3S when these databases also identified one of the common storms. Storm XYNTHIA (2010-02-27)
illustrates this well, as XYNTHIA is least severe as derived from ERA5, much stronger as derived from CCLM_ERA5_EUR-11 and XWS, and strongest as derived from COSMO-REA6 and CCLM_ERA5_CEU-3, which exhibit similar normalized losses. Storm KYRILL (2007-01-19) is instead an example of relatively good agreement among data sets, as it was the strongest storm within our database and C3S in terms of ordinal rank (Table 2) and normalized losses (Figure 6), and it was the second-strongest storm within XWS.

No clear overarching trends in storm severity with data set emerge in Figure 6, but some tendencies are apparent. The common storms tend to be stronger as derived from ERA5 and COSMO-REA6, in comparison to CCLM_ERA5_EUR-11 and CCLM_ERA5_CEU-3, as implied by Figure 6 and indicated by the the median and 75$^{th}$ percentile values of the common storm normalized losses for each input data set (Table 5). CCLM_ERA5_CEU-3, in turn, tends to exhibit the weakest storms, with smaller median, 25$^{th}$ percentile, and 75$^{th}$ percentile values than most of the other three input sources. However, the interquartile
ranges for the input data sets in Table 5 overlap with each other. Table 5 also demonstrates that the median and 25$^{th}$ percentile relative rank values for XWS fall within the ranges exhibited by our database, while the 75th percentile is larger than in any of the four data sets we use. C3S often exhibits the strongest or near-strongest common storms relative to our database and to XWS (Figure 5). Its 25$^{th}$ percentile value is larger than all medians in Table 5 with the exception of COSMO-REA6, suggesting statistically significantly larger storm severities compared to our database and XWS.

The substantial variation in storm severity with data set exhibited by the common storms, both within our database and among the three databases, point to the impact of horizontal resolution and storm identification methodology on storm identification





and estimated severity. Additional causes, however, are likely also at play, such as the influence of the configuration of the input data sets themselves on their representations of extreme storms. Because our database includes four different input data sets, this affords a unique perspective to investigate the cause(s) of these variations and their influence on extreme storm characterization and impacts prediction.

## 3.4 Spatial Variability among Data Sets

### 3.4.1 Spatial Variability of Storm Footprints

For the spatial comparison of the common storms, the ERA5, CCLM_ERA5_EUR-11, and COSMO-REA6 footprints and corresponding absolute wind gusts, and the XWS and C3S absolute wind gusts, were regridded using bilinear interpolation to the CCLM_ERA5_CEU-3 horizontal resolution and restricted to the enlarged Germany domain. The mean of the footprint differences over all 29 common storms was then taken for each input data set and are presented as the mean footprint differences in Figure 7; the same was done with the absolute wind gusts and the mean absolute wind gust differences are presented in Figure 8. As these differences were not computed as absolute differences, the means presented in Figures 7 and 8 carry the risk of the cancellation of errors, which is particularly important for interpretation of mean differences of or near zero. Comparisons were made to CCLM_ERA5_CEU-3 because this input data set has the highest native resolution and smallest native domain; CCLM_ERA5_CEU-3 is not assumed to be the most accurate or realistic data set. This analysis is not intended to evaluate which input source is the most accurate, but rather seeks to characterize the main spatial differences arising among the input data sets used in our database.

The footprint comparison between ERA5 and CCLM_ERA5_CEU-3 stands out, as it displays primarily positive mean differences over the enlarged Germany domain with few exceptions, indicating that, on average, the footprint as derived from CCLM_ERA5_CEU-3 is larger in magnitude than as derived from ERA5 (Figure 7). The footprint comparisons between CCLM_ERA5_EUR-11 or COSMO-REA6 and CCLM_ERA5_CEU-3 are more variable in terms of positive or negative mean difference over this domain. However, the mean footprint differences between CCLM_ERA5_CEU-3 and COSMO-REA6 tend to exhibit more locations with negative sign or smaller-magnitude positive differences than do the differences with respect to CCLM_ERA5_EUR-11. The footprint comparison between ERA5 and CCLM_ERA5_CEU-3 also stands out in terms of the larger magnitude of the differences, regardless of sign. Many locations within the domain display a footprint difference magnitude between 0.2 and 0.3, while the smaller cluster of locations exhibiting larger-magnitude negative differences within Czechia and Slovakia can reach a magnitude between -0.2 to -0.4. This may be partially due to ERA5 having the coarsest original native horizontal resolution compared to CCLM_ERA5_CEU-3. In comparison, the differences with CCLM_ERA5_EUR-11 and COSMO-REA6 tend to display magnitudes between -0.2 and 0.2 over most of the enlarged Germany domain.

These comparisons thus reveal substantial disagreements among the input data sets in the spatial structures of the common storm footprints, though only the comparison between CCLM_ERA5_CEU-3 and ERA5 presents a domain-scale systematic difference. However, the mean differences can conceal other important disagreements among the footprints; these will be discussed in Section 3.4.3.



### 3.4.2 Spatial Variability of Associated Absolute Wind Gusts

Expectedly, the mean differences in the absolute wind gusts associated with the common storm footprints also reveal substantial disagreements among the input data sets, and with XWS and C3S (Figure 8). The mean footprint differences and mean absolute wind gust differences relative to CCLM_ERA5_CEU-3 for each input data set are similar to each other in an overall sense (e.g., a positive, larger magnitude difference in absolute wind gust in a region exhibiting a positive, larger magnitude difference in the footprint). However, locations with differences of opposing sign between the footprint and absolute wind gusts are also apparent, as are locations where the difference magnitudes are larger for the footprint but smaller for the absolute wind gusts, and vice versa. This seeming contradiction may be most readily seen in the comparison to ERA5. For example, the mean footprint differences between CCLM_ERA5_CEU-3 and ERA5 over northwestern Germany and the Netherlands are generally small and positive (between 0.0 and 0.1), and the mean absolute wind gust differences are close to zero or more substantially negative (approximately -2.2 ms$^{-1}$ or more negative). The mean absolute wind gust differences for CCLM_ERA5_EUR-11 exhibit broader regions of positive difference (0.0 to approximately 4.4 ms$^{-1}$ or greater) over much of continental Europe, in contrast to the mean footprint differences of small magnitude and variable sign. The mean footprint and absolute wind gust comparisons most resemble each other over the northern portion of the domain (North and Baltic Seas, Denmark, and southern Sweden) for each input data set. ERA5 no longer stands out over much of continental Europe as displaying larger magnitude differences compared to the other two data sets when comparing the absolute wind gusts.

The mean footprint and absolute wind gust comparisons between CCLM_ERA5_CEU-3 and COSMO-REA6 resemble each other more than is the case for the other two input data sets, as would be expected since they are derived from a common numerical model. The differences identified between all data sets in terms of wind gusts could arise, for example, from different boundary conditions, different wind gust computations or parameterizations, or configuration options unique to a data set. Increasing horizontal resolution can also allow for greater spatial heterogeneity within a data set's wind gust field, and thus greater heterogeneity within the the threshold exceedances used to compute the footprints.

The comparisons for XWS or C3S further reflect impacts on the wind gust fields within these databases arising from different horizontal resolutions, the use of ERA-Interim or ERA5 to create the database, and the choice of storm identification methodology, and partially reflect that not all common storms were captured by these two databases. The XWS and C3S comparisons do not resemble each other, nor do they resemble the comparisons within our own database. For example, the comparison to C3S displays quite substantially positive differences over almost the entire eastern half of the domain not seen in the other data sets; these differences are much larger in magnitude than the positive differences exhibited by the comparison to CCLM_ERA5_EUR-11, and lack the regions of very small to no difference that are also seen in the CCLM_ERA5_EUR-11 comparison. The comparison to XWS is more variable in sign and magnitude over the central continental portion of the domain, but displays substantially negative differences over Denmark, southern Sweden, Poland, and the Baltic Sea that are in stark contrast to the remaining data sets. The comparison to XWS indeed exhibits the broadest area of negative difference of any of the data sets, indicating that the absolute wind gusts from XWS tend to be larger than those from CCLM_ERA5_CEU-3.





### 3.4.3 Windstorms KYRILL and ANDREA as Comparative Examples

As mentioned above, individual storm footprints and absolute wind gusts contain manifestations of spatial variability across
data sets additional to those apparent in the mean comparisons. The comparisons between CCLM_ERA5_CEU-3 and the
remaining data sets for the individual storms KYRILL (2007-01-18) and ANDREA (2012-01-05) provide such examples.
KYRILL is the strongest storm within our database and C3S, and the second-strongest within XWS. ANDREA also falls
within the top 10 most extreme storms for all data sets except XWS (Table 2), but, unlike KYRILL, its relative storm strengths
are much more variable among input sources (Figure 6). Most footprint and absolute wind gust comparisons for KYRILL
(Figures 9 and 10) and ANDREA (Figures 11 and 12) depart substantially from the mean comparisons. The ordinal and
relative strengths for KYRILL and ANDREA serve to highlight that agreement in spatial variability among input data sets
does not depend on storm severity, nor does it depend on agreement on storm severity among input data sets.

The KYRILL and ANDREA CCLM_ERA5_CEU-3 footprints display a signature of convective activity over continental
Europe, albeit stronger in KYRILL, that is weaker or missing in the other data sets. This appears as the "straight line" maximum
relative wind gusts apparent in Figures 1, 9, and 11. High resolution and a convection-permitting configuration, such as in
CCLM_ERA5_CEU-3, contribute to better representation of the downdrafts and wind gusts arising from convective activity
associated with winter windstorms, causing the "straight line" relative wind gusts in the footprint (Ludwig et al., 2015). Several
storms in our database exhibit this signature in CCLM_ERA5_CEU-3 and occasionally other input data sets. ERA5 typically
misses it entirely, with an increasing ability to represent such mesoscale processes for CCLM_ERA5_EUR-11 and COSMO-
REA6, though not as well as CCLM_ERA5_CEU-3. Spatial variability arising from convective activity is entirely missing in
the mean footprint comparison.

The mean comparisons also conceal the differences in footprint spatial extents or boundaries among data sets for the same
storm, as appears in the KYRILL and ANDREA comparisons. Disagreements in extent typically occur in the eastern portion
of and the southern boundary of the domain, as highlighted by the bright red regions for KYRILL and ANDREA indicating
that CCLM_ERA5_CEU-3 locates the footprint there but the other data set did not (Figures 9 and 11). In addition to footprint
boundary disagreements, gaps within footprints containing locations that were not impacted by the surrounding storm exist
for some data sets but not others. Such a gap is apparent in the comparison between CCLM_ERA5_CEU-3 and ERA5 for
ANDREA over the Danish islands, Baltic Sea, and northeastern Germany (Figure 11). The causes for disagreements in footprint
extents and locations included in a footprint interior are unclear, but again likely related to horizontal resolution, footprint
identification methodology, and data set configuration differences. Care should thus be taken when choosing an appropriate
footprint source(s) for loss estimations.

## 4 Conclusions

We have presented and characterized a new database of the Top50 most extreme European winter windstorms over the 1995-
2015 extended winter seasons based on a consistent footprint identification and severity assessment methodology. We used four
different meteorological input data sets, namely the ERA5, CCLM_ERA5_EUR-11, COSMO-REA6, and CCLM_ERA5_CEU-



3 reanalyses or regional climate model simulation outputs. We applied the same footprint methodology to each input data set on its native horizontal resolution and spatial domain, thereby creating a consistent, systematic database of the most extreme storms over a 20-year time record identified across diverse input data sources. This allows for a direct comparison of the effects of using different wind gust input data sources on the storm footprints, their severity, and their spatial characteristics. Equally importantly, the temporal range of the data sets can be easily extended when new data become available, and it is equally easy for users to include additional input data sets to which they may have access. Our database therefore provides complementary perspectives on extreme storm identification and severity assessment given the diverse horizontal resolutions and input sources used, and complements existing extreme European winter windstorm databases.

Our results highlight the major effects that horizontal resolution, choice of footprint identification methodology, and differing types of input data sets can have on extreme storm identification and characterization. We found variability across input data sets in all aspects of our database, from which storms were identified as belonging to the Top50 storms to the quite substantial variability in the spatial structure of the footprints for storms identified in more than one input data set, as well as between our database and the XWS and C3S databases.

Our database identified a total of 76 extreme storms, with 29 common storms identified in all four input sources. Another 47 storms were identified in at least one of the input sources but not all four. The strongest storms, such as KYRILL, LOTHAR, and JEANETT, tended to be identified in all four input sources, though the common storms also include many comparatively weaker storms. However, the severity of a given storm, as denoted by a storm's ordinal rank and normalized loss, often varied across the input data sets in which it was identified for both the common storms and the storms identified in only two or three inputs. The variation in relative storm strengths could be substantial across input data sets for many storms, though the storms as identified by C3S tended to be stronger than as identified in the other data sets. The input data sets, XWS, and C3S disagreed on the winters with low and high storm activity, and, while a large, statistically significant positive correlation exists between the proportion of all storms per winter and the proportion of total losses per winter for each input data set, they disagree on how relatively damaging each winter was. Equally importantly, the footprints and associated absolute wind gusts themselves exhibited substantial variability across input data sets and with XWS and C3S, both in the mean comparisons and for individual storms. Comparisons of the footprint and absolute wind gust spatial variability for individual storms revealed smaller scale features not present in the mean comparisons, such as the signatures of convective activity. Mild disagreement of one day is also seen for some storms in their dates of occurrence, but the majority of storms within our database agreed on their dates of occurrence.

We underscore that our goal in creating this database was not to provide a record of accurate and inaccurate footprints nor to assess which input data set or horizontal resolution performed best at identification and characterization of extreme European winter storms. Rather, we seek to provide a diverse yet consistent record of storm footprints that allows for different perspectives on these storms. A diversity of perspectives has been lacking for footprint data, and thus, by providing multiple sources but a consistent methodology, our database supports uncertainty quantification for a deeper understanding of extreme storms and their impacts. Indeed, uncertainty quantification for footprint data is a little-investigated topic, and our database could allow for assignment of an uncertainty range in the footprints, or the design and assessment of another uncertainty metric

after the needs of specific end-users. Combining analysis of the uncertainties associated with the footprints and storm losses may lead to improved models of storm damages and improved understanding of how such storms may change with climate change. Storm footprints derived from multiple data sources on different resolutions could also present an advantage for many areas of research that currently give little weight to historical data uncertainty, such as storm clustering studies.

It is likely that the footprint(s) that are most useful to a particular user of our database could depend heavily on that user's specific goals; for example, interest in a storm affecting a mountainous region might call for footprints at higher horizontal resolution and neglect of those at coarser resolution. However, based on our comparisons here, it is most useful to consider the storms as identified from two or more input sources within our database, where possible, rather than a footprint from a single source. We exhort users to do so, in order to provide information on the uncertainties associated with the storms and
their impacts and more reliable analysis.

## 5   Data availability

The new storm database we developed and present here is publicly available online through Zenodo at https://doi.org/10.5281/zenodo.10594399 (Flynn et al., 2024). The ERA5 reanalysis data and the C3S winter windstorm indicators for Europe (the C3S windstorm database), used as an input source for our database and as a comparison to our database, re-
spectively, are publicly available online through the Copernicus Climate Change Service (https://cds.climate.copernicus.eu/). The CCLM_ERA5_CEU-3 simulation output (Brienen et al., 2022), used as an input source for our database, is publicly available online from the DWD Earth System Federation Grid nodes (https://esgf.dwd.de/search/esgf-dwd/). The COSMO-REA6 data set, used as an input source for our database, is publicly available from DWD (https://opendata.dwd.de/climate_environment/REA/). The XWS database, used as a comparison to our database, is publicly available online (https://www.
europeanwindstorms.org/repository/).

*Author contributions.* GM and JGP conceived of the database. CMF performed the data collection and analysis and wrote the initial paper draft. JM provided guidance to CMF in how to compute the storm footprints and loss indices, and how to rank the storms. All authors discussed the results and contributed to drafting the manuscript.

*Competing interests.* The authors declare that they have no conflicts of interest.

*Acknowledgements.* This research has received funding from the European Union's Horizon research and innovation programme under European Research Council grant no. 101112727. JM was funded by the Bundesministerium für Bildung und Forschung (BMBF; German Ministry for Education and Research) under project "RegIKlim-NUKLEUS" (01LR2002B1). JGP thanks the AXA Research Fund for support.



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





**Table 1.** Summary of input data sets used to create the extreme windstorms database, or used for comparison to the database.

| Data Set Name | Data Set Type | Domain | Domain Boundaries | Horizontal Resolution | Number of Storms Identified |
|---|---|---|---|---|---|
| ERA5 | Global Reanalysis | Global (European sub-domain selected) | 27 °N–72 °N, 22 °W–45 °E | 0.25 ° | 50 |
| CCLM_ERA5_EUR-11 | Regional Climate Model | CORDEX EUR-11 | ∼ 27 °N–72 °N, ∼ 22 °W–45 °E | 0.11° (∼ 12 km) | 50 |
| COSMO-REA6 | Regional Reanalysis | CORDEX EUR-11 | ∼ 27 °N–72 °N, ∼ 22 °W–45 °E | 0.055° (∼ 6 km) | 50 |
| CCLM_ERA5_CEU-3 | Regional Climate Model | enlarged Germany domain | ∼ 45.5 °N–58 °N, ∼ 0.9 °W–20 °E | 0.0275° (∼ 2.8 km) | 50 |
| XWS | Derived from ERA-Interim Reanalysis and insurance data | Europe | ∼ 36 °N–68 °N, ∼ 20 °W–40 °E | ∼ 25 km | 23 |
| C3S | Derived from statistical downscaling of ERA5 | Europe | ∼ 30 °N–70 °N, ∼ 25 °W–40 °E | 1 km | 30 |



**Table 2.** Summary of the Top50 extreme windstorms common to all of the four input data sets, with their dates of occurrence and ordinal ranks (rank 1 = most extreme storm, rank 50 = least extreme storm of the Top50 storms). A storm name in parentheses indicates that the same storm had two different names, while a hyphenated name indicates that two individual storms could not be effectively separated from each other within the database. For storms lacking given names and thus named after their date of occurrence, the ERA5 date was preferred.

| Storm Name | ERA5 Date | Rank | CCLM_ERA5_EUR-11 Date | Rank | COSMO-REA6 Date | Rank | CCLM_ERA5_CEU-3 Date | Rank | XWS Date | Rank | C3S Date | Rank |
|---|---|---|---|---|---|---|---|---|---|---|---|---|
| ANATOL | 1999-12-03 | 18 | 1999-12-03 | 20 | 1999-12-03 | 14 | 1999-12-03 | 19 | 1999-12-03 | 3 | 1999-12-03 | 16 |
| ANDREA (ULLI) | 2012-01-05 | 4 | 2012-01-05 | 5 | 2012-01-05 | 5 | 2012-01-05 | 8 | 2012-01-03 | 12 | 2012-01-03 | 4 |
| ANNA | 2002-02-26 | 24 | 2002-02-26 | 17 | 2002-02-26 | 19 | 2002-02-26 | 25 | X | X | 2002-02-26 | 19 |
| ARIANE | 1997-02-13 | 32 | 1997-02-13 | 12 | 1997-02-13 | 10 | 1997-02-14 | 18 | X | X | X | X |
| CARMEN | 2010-11-12 | 33 | 2010-11-12 | 43 | 2010-11-12 | 48 | 2010-11-12 | 43 | X | X | X | X |
| ELIVRA-FARAH | 1998-03-04 | 20 | 1998-03-04 | 25 | 1998-03-04 | 22 | 1998-03-04 | 24 | X | X | X | X |
| EMMA | 2008-03-01 | 6 | 2008-03-01 | 19 | 2008-03-01 | 8 | 2008-03-01 | 4 | 2008-02-29 | 21 | 2008-03-02 | 6 |
| FANNY | 1998-01-04 | 12 | 1998-01-04 | 11 | 1998-01-04 | 11 | 1998-01-04 | 27 | 1998-01-04 | 20 | 1998-01-04 | 11 |
| FRANZ | 2007-01-11 | 7 | 2007-01-12 | 15 | 2007-01-11 | 13 | 2007-01-11 | 12 | X | X | X | X |
| FRIDTJOF | 2007-12-02 | 23 | 2007-12-02 | 23 | 2007-12-02 | 28 | 2007-12-02 | 37 | X | X | X | X |
| GISELA-HEIDI | 1997-02-25 | 9 | 1997-02-25 | 16 | 1997-02-25 | 9 | 1997-02-25 | 14 | X | X | 1997-02-23 | 8 |
| GUNTER | 2015-01-10 | 22 | 2015-01-10 | 21 | 2015-01-10 | 24 | 2015-01-10 | 16 | X | X | 2015-01-10 | 18 |
| ILONA | 2002-01-27 | 26 | 2002-01-27 | 28 | 2002-01-27 | 25 | 2002-01-27 | 17 | X | X | 2002-01-27 | 15 |
| JEANETT | 2002-10-27 | 5 | 2002-10-27 | 4 | 2002-10-27 | 3 | 2002-10-27 | 2 | 2002-10-27 | 1 | 2002-10-27 | 5 |
| JOACHIM | 2011-12-16 | 30 | 2011-12-16 | 31 | 2011-12-16 | 30 | 2011-12-16 | 31 | 2011-12-16 | 9 | 2011-12-16 | 23 |
| KERSTIN-LIANE | 2000-01-30 | 43 | 2000-01-30 | 44 | 2000-01-30 | 39 | 2000-01-30 | 33 | X | X | X | X |
| KIRSTEN | 2008-03-11 | 11 | 2008-03-11 | 7 | 2008-03-11 | 4 | 2008-03-11 | 13 | X | X | 2008-03-11 | 10 |
| KYRILL | 2007-01-18 | 1 | 2007-01-18 | 1 | 2007-01-18 | 1 | 2007-01-18 | 1 | 2007-01-18 | 2 | 2007-01-18 | 1 |
| LARA | 1999-02-05 | 40 | 1999-02-05 | 32 | 1999-02-05 | 33 | 1999-02-05 | 23 | X | X | X | X |
| LOTHAR | 1999-12-26 | 2 | 1999-12-26 | 3 | 1999-12-26 | 2 | 1999-12-26 | 3 | 1999-12-26 | 13 | 1999-12-26 | 2 |
| MIKE-NIKLAS | 2015-03-30 | 14 | 2015-04-01 | 10 | 2015-03-30 | 6 | 2015-30-30 | 6 | X | X | X | X |
| NILS | 2015-11-29 | 37 | 2015-11-29 | 50 | 2015-11-29 | 47 | 2015-11-30 | 38 | X | X | X | X |
| NINA-ORALIE | 2004-03-20 | 29 | 2004-03-20 | 36 | 2004-03-20 | 18 | 2004-03-20 | 26 | X | X | X | X |
| ORATIA | 2000-10-30 | 10 | 2000-10-30 | 22 | 2000-10-30 | 7 | 2000-10-30 | 42 | 2000-10-30 | 4 | 2000-10-30 | 9 |
| u19950216 | 1995-02-16 | 47 | 1995-02-17 | 47 | 1995-02-17 | 37 | 1995-02-16 | 46 | X | X | X | X |
| ULF | 2005-02-13 | 48 | 2005-02-13 | 24 | 2005-02-13 | 35 | 2005-02-13 | 28 | X | X | X | X |
| XAVER | 2013-12-05 | 28 | 2013-12-05 | 26 | 2013-12-05 | 20 | 2013-12-05 | 20 | 2013-12-05 | 6 | 2013-12-05 | 22 |
| XYLIA | 1998-10-28 | 16 | 1998-10-28 | 9 | 1998-10-28 | 12 | 1998-10-28 | 10 | 1998-10-28 | 23 | 1998-10-28 | 14 |
| XYNTHIA | 2010-02-27 | 45 | 2010-02-27 | 29 | 2010-02-28 | 15 | 2010-02-28 | 11 | 2010-02-27 | 7 | X | X |



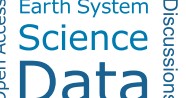

**Table 3.** Summary of the winters within each input data set, XWS, and C3S that belong to the periods of no storm activity (0% of all storms during one winter season), low storm activity (2% of all storms during one winter season), high storm activity (8% of all storms during one winter season), and very high storm activity (10+% of all storms during one winter season). The percentages of total storm losses that occurred during each winter for each data set are given in parentheses. Winters are listed in chronological order for each data set and section.

| | ERA5 (%) | CCLM_ERA5_EUR-11 (%) | COSMO-REA6 (%) | CCLM_ERA5_CEU-3 (%) | XWS (%) | C3S (%) |
|---|---|---|---|---|---|---|
| No Storm Activity | 1995/1996 (0%) | 1995/1996 (0%) | 1995/1996 (0%) | 1995/1996 (0%) | 1995/1996 (0%) | 1995/1996 (0%) |
| (0% of all storms) | 2005/2006 (0%) | 2005/2006 (0%) | 2005/2006 (0%) | 2012/2013 (0%) | 2003/2004 (0%) | 2003/2004 (0%) |
| | 2008/2009 (0%) | 2008/2009 (0%) | 2008/2009 (0%) | | 2005/2006 (0%) | 2005/2006 (0%) |
| | 2012/2013 (0%) | 2012/2013 (0%) | 2012/2013 (0%) | | 2010/2011 (0%) | 2009/2010 (0%) |
| | | | | | 2012/2013 (0%) | 2010/2011 (0%) |
| | | | | | 2014/2015 (0%) | 2012/2013 (0%) |
| | | | | | 2015/2016 (0%) | 2015/2016 (0%) |
| Low Storm Activity | 2003/2004 (1.8%) | 2000/2001 (1.9%) | 2002/2003 (3.4%) | 2000/2001 (1.5%) | | |
| (2% of all storms) | 2009/2010 (1.3%) | 2002/2003 (3.2%) | 2004/2005 (1.6%) | 2005/2006 (1.7%) | | |
| | 2015/2016 (1.4%) | 2010/2011 (1.5%) | 2010/2011 (1.4%) | 2008/2009 (1.6%) | | |
| | | 2015/2016 (1.3%) | | 2009/2010 (2.3%) | | |
| | | | | 2010/2011 (1.4%) | | |
| High Storm Activity | 1994/1995 (6.9%) | 2006/2007 (9.2%) | 1994/1995 (7.3%) | 1996/1997 (8.1%) | 1997/1998 (4.5%) | |
| (8% of all storms) | 1998/1999 (6.8%) | 2007/2008 (7.9%) | 1996/1997 (8.4%) | | 1998/1999 (4.4%) | |
| | 2006/2007 (10.2%) | 2013/2014 (7.9%) | 2003/2004 (6.4%) | | 2011/2012 (7.2%) | |
| | 2007/2008 (8.6%) | | 2007/2008 (8.7%) | | 2013/2014 (7.2%) | |
| | 2013/2014 (7.9%) | | | | | |
| Very High Storm Activity | 1999/2000 (11.6%) | 1996/1997 (11.4%) | 2006/2007 (10.6%) | 1999/2000 (10.1%) | 1996/1997 (8.9%) | 1996/1997 (10.3%) |
| (10+% of all storms) | | 1999/2000 (13.9%) | 2013/2014 (9.0%) | 2003/2004 (7.9%) | 1999/2000 (15.2%) | 2007/2008 (9.3%) |
| | | | | 2004/2005 (8.7%) | | 2013/2014 (11.8%) |





**Table 4.** Summary of the mean difference (±1 standard deviation) between XWS or C3S and the input data sets within our database for the proportion of all storms per winter and the proportion of total storm losses per winter. Differences are computed as the absolute value of the difference between XWS or C3S and the input data sets for each winter.

|  |  | ERA5 | CCLM_ERA5_EUR-11 | COSMO-REA6 | CCLM_ERA5_CEU-3 |
|---|---|---|---|---|---|
| Proportion of All Storms | XWS | 2.24% (±1.85%) | 2.24% (±1.78%) | 3.08% (±2.29%) | 3.11% (±2.28%) |
| Per Winter | C3S | 2.48% (±1.99%) | 2.56% (±1.88%) | 2.89% (±2.34%) | 3.18% (±2.88%) |
| Proportion of Total Losses | XWS | 2.54% (±2.14%) | 2.27% (±2.38%) | 2.88% (±2.81%) | 2.86% (±2.28%) |
| Per Winter | C3S | 1.83% (±1.45%) | 1.89% (±1.36%) | 2.31% (±2.19%) | 2.69% (±2.38%) |



**Table 5.** Summary of the 25th percentile value, median value, and 75th percentile value of the normalized losses (relative ranks) over all the common storms for each input data set, and over the common storms available from XWS and C3S. All quantities are unitless.

|  | ERA5 | CCLM_ERA5_EUR-11 | COSMO-REA6 | CCLM_ERA5_CEU-3 | XWS | C3S |
|---|---|---|---|---|---|---|
| 25th Percentile | 0.15 | 0.18 | 0.19 | 0.16 | 0.17 | 0.36 |
| Median | 0.31 | 0.27 | 0.37 | 0.24 | 0.24 | 0.44 |
| 75th Percentile | 0.46 | 0.39 | 0.47 | 0.31 | 0.48 | 0.55 |

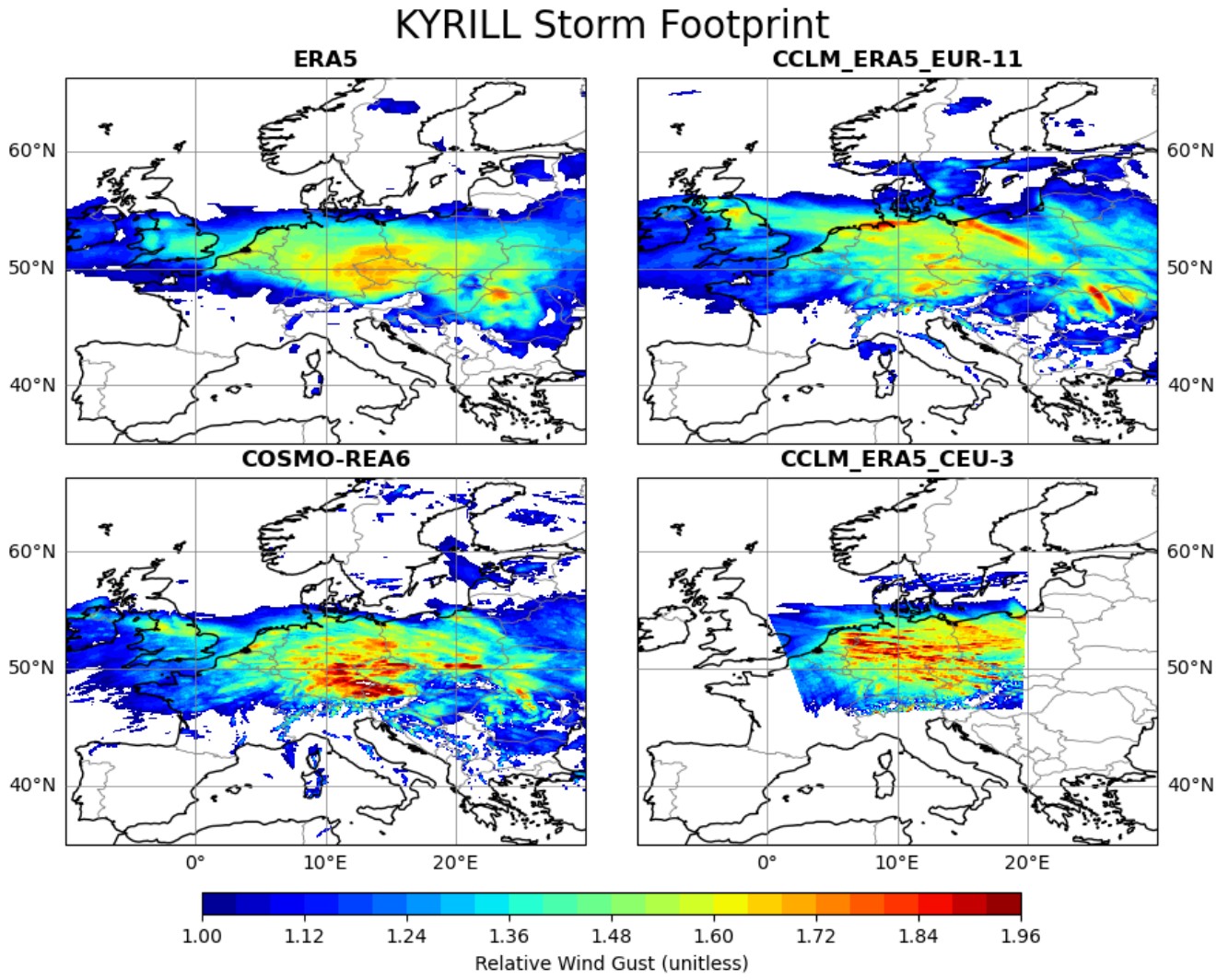

**Figure 1.** Storm footprints, as represented by the locations for which the relative wind gust was greater than 1.0, for windstorm KYRILL (Jan 2007) for the four input data sets used here. For XWS and C3S it was not possible to compute or obtain the relative wind gusts. All footprints are plotted at their native horizontal resolution and for their native domain.



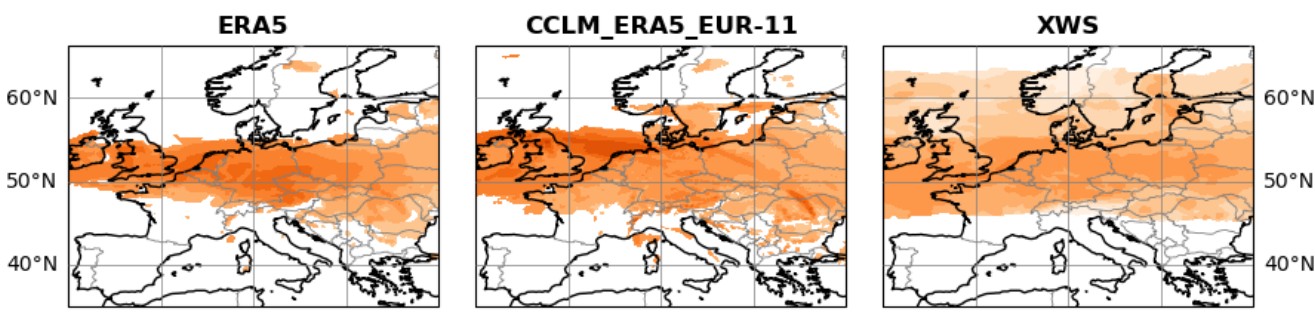

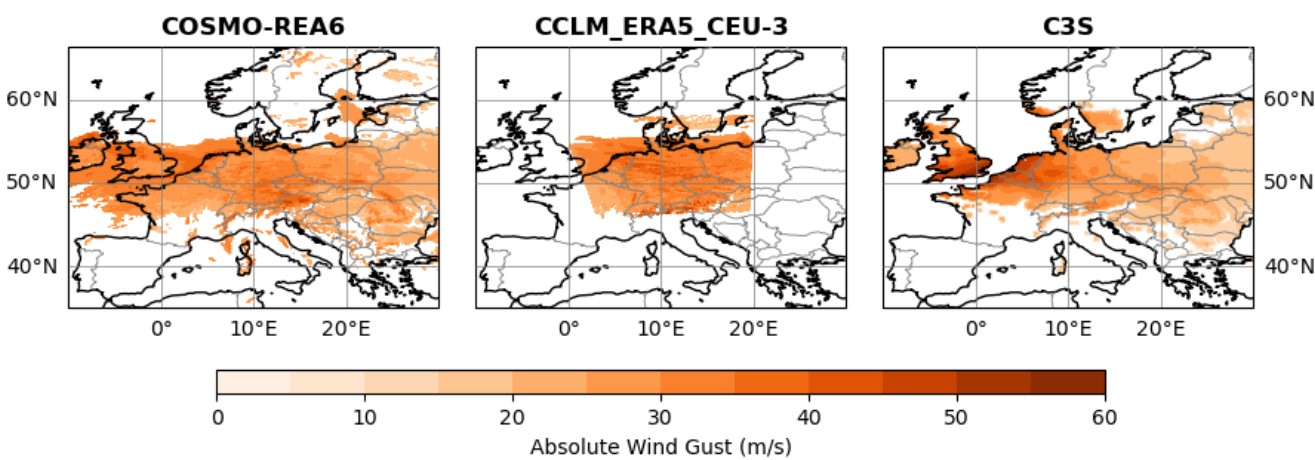

**Figure 2.** The daily maximum wind gusts (m s$^{-1}$) for windstorm KYRILL (Jan 2007) at each location associated with the storm footprints displayed in Fig. 1, or which were provided by the XWS and C3S databases. All absolute wind gusts are plotted at their native horizontal resolution and for their native domain.

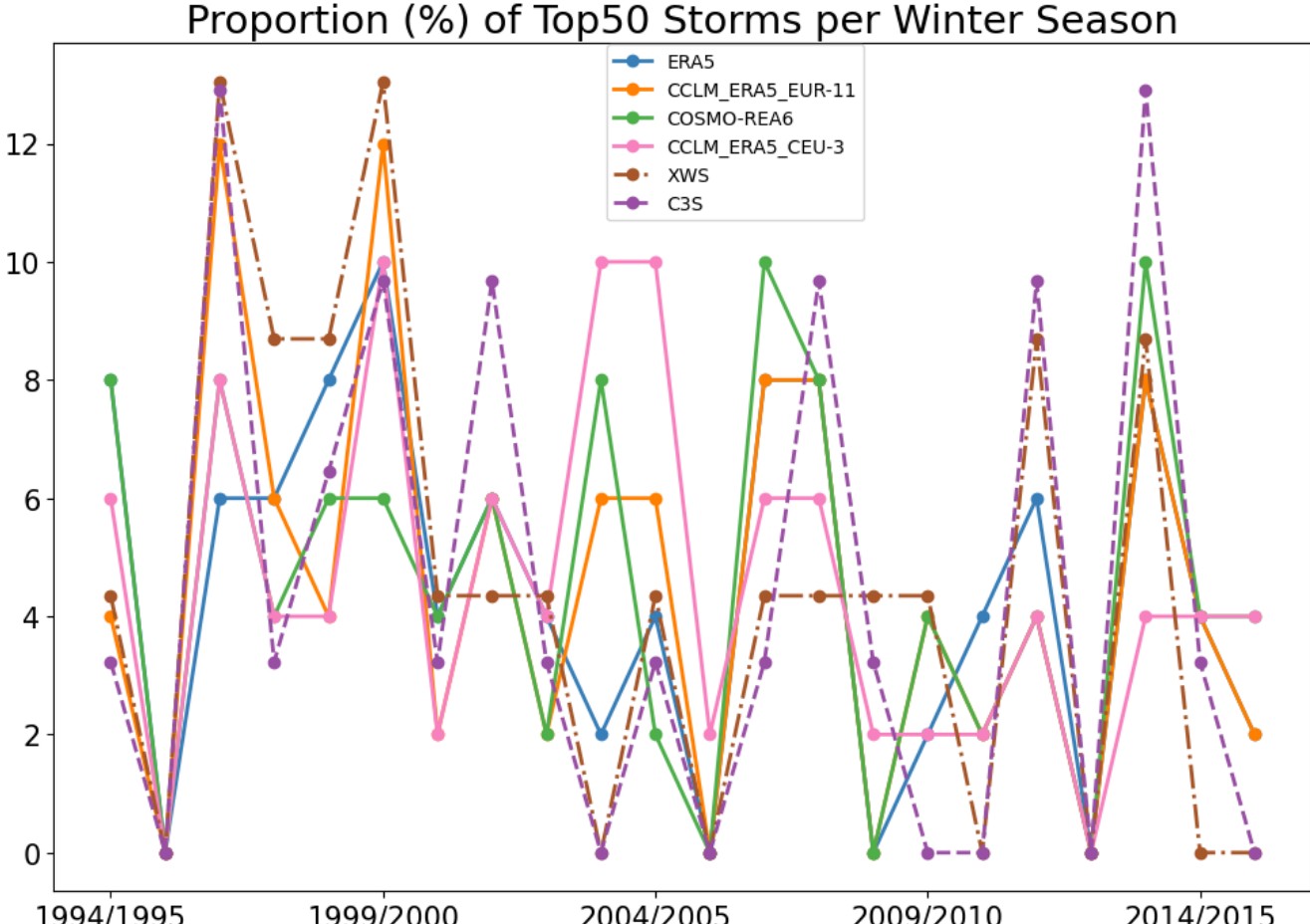

**Figure 3.** The percentage of extreme storms (y-axis) that occurred during each extended winter season (ONDJFM; x-axis) for each input data set (solid colored lines), and XWS and C3S (dashed colored lines) over the 20-year period covering the 1994/1995 - 2015/2016 extended winter seasons. The end-cap extended winter seasons 1994/1995 and 2015/2016 exclude storms that occurred in the years 1994 and 2016. Percentages computed relative to the total number of extreme storms for each data set: 50 storms for ERA5, CCLM_ERA5_EUR-11, COSMO-REA6, and CCLM_ERA5_CEU-3; 23 for XWS; and 30 for C3S.

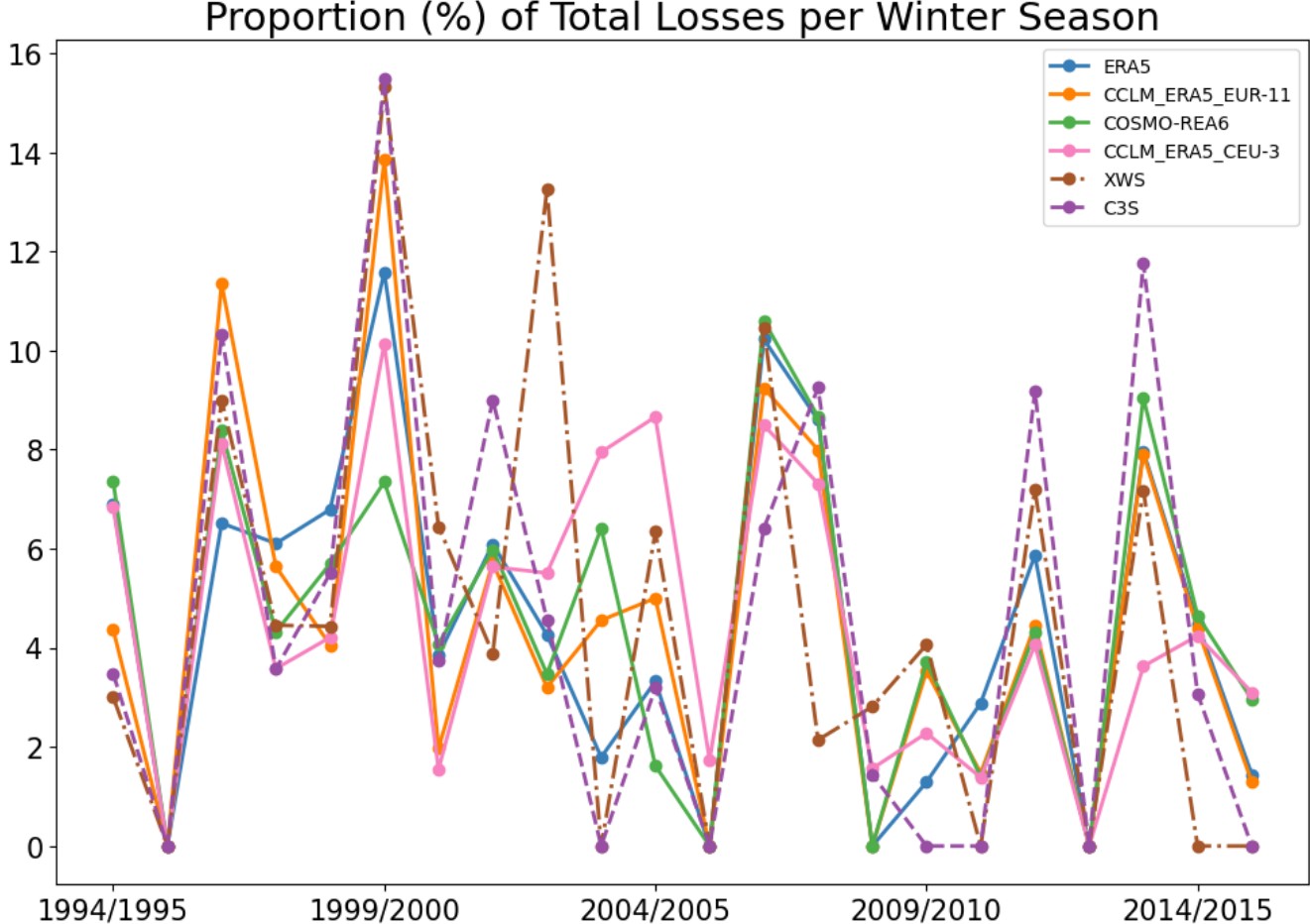

**Figure 4.** The percentage of total normalized loss (y-axis) that was incurred per extended winter season (ONDJFM; x-axis) for each input data set (solid colored lines), and XWS and C3S (dashed colored lines) over the 20-year period covering the 1994/1995 - 2015/2016 extended winter seasons. The end-cap extended winter seasons 1994/1995 and 2015/2016 exclude storms that occurred in the years 1994 and 2016.





**Figure 5.** The OEP/AEP ratio (y-axis; unitless) per extended winter season (ONDJFM; x-axis) for each input data set (solid colored lines), and XWS and C3S (dashed colored lines) over the 20-year period covering the 1994/1995 - 2015/2016 extended winter seasons. The ratio varies between undefined for winters with no storm activity to a maximum of 1.0 for winters containing only a single storm. The end-cap extended winter seasons 1994/1995 and 2015/2016 exclude storms that occurred in the years 1994 and 2016.

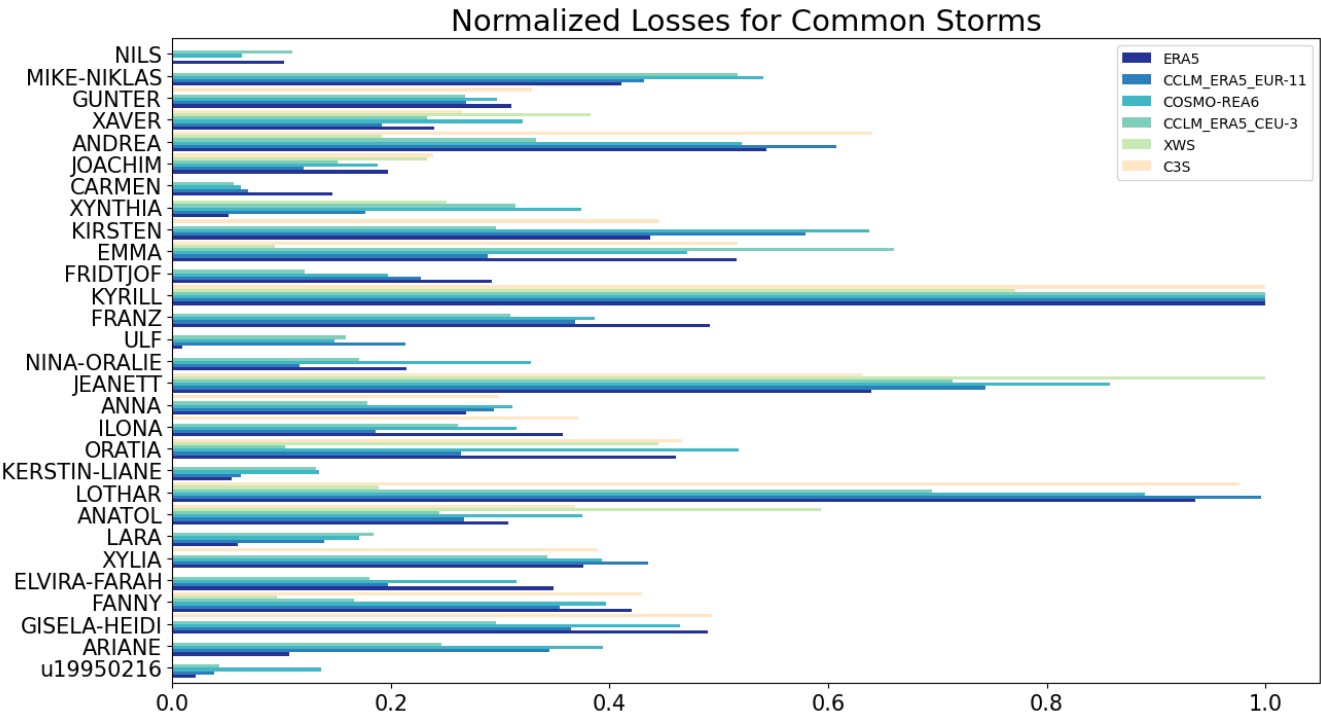

**Figure 6.** Bar plots indicating the normalized loss (x-axis; unitless) per common storm (y-axis) for each of the input data sets, and for XWS and C3S when available. Storms are sorted into chronological order, with the most recent storms at the top and least recent at the bottom. Common storms only.







**Figure 7.** Mean storm footprint difference computed over the common storms only between CCLM_ERA5_CEU-3 and, in clockwise order, the ERA5, CCLM_ERA5_EUR-11, and COSMO-REA6 data sets. The colors represent the mean difference in the relative wind (unitless). All panels are plotted at the CCLM_ERA5_CEU-3 horizontal resolution and on the enlarged Germany domain.




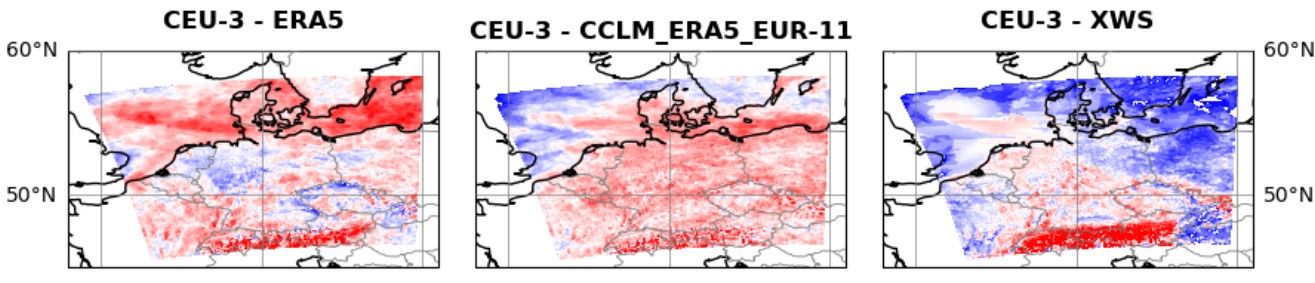

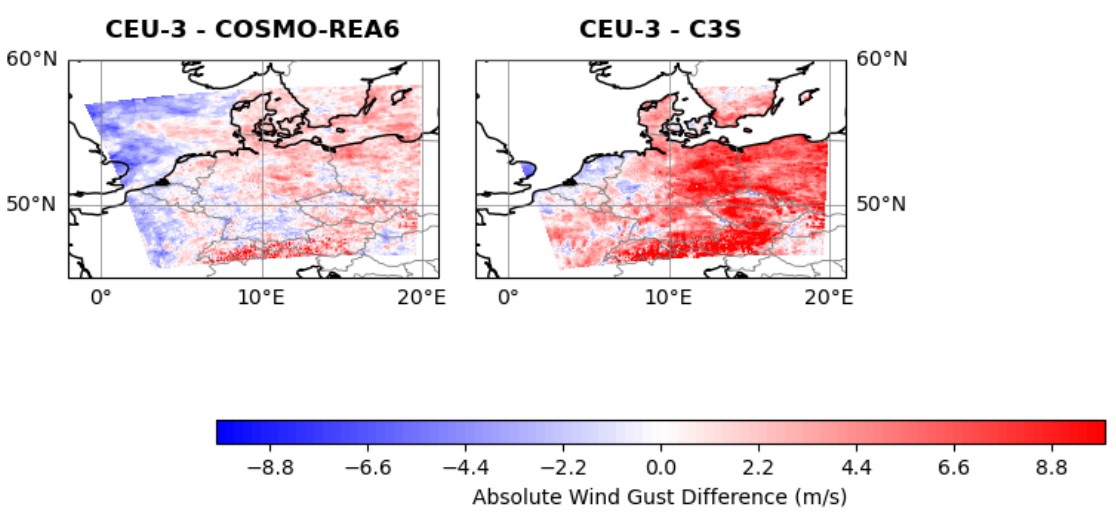

**Figure 8.** Mean difference in daily maximum wind gusts associated with the footprints computed over the common storms only between CCLM_ERA5_CEU-3 and, in clockwise order, the ERA5, CCLM_ERA5_EUR-11, XWS, C3S, and COSMO-REA6 data sets. The colors represent the mean difference in the absolute wind gusts (ms$^{-1}$). All panels are plotted at the CCLM_ERA5_CEU-3 horizontal resolution and on the enlarged Germany domain.









**Figure 9.** Storm footprint for windstorm KYRILL (Jan 2007) derived from the CCLM_ERA5_CEU-3 data set and shown in Fig. 1, displayed in the top left panel. The three remaining panels display the difference between the CCLM_ERA5_CEU-3 footprint and the footprints derived from ERA5, COSMO-REA6, and CCLM_ERA5_EUR-11 after regridding to the CCLM_ERA5_CEU-3 resolution, in clockwise order. Colors for the CCLM_ERA5_CEU-3 footprint are the same as in Figure 1, while the the colors representing the footprint differences displayed in the remaining three panels are defined by the color bar below the plot. All panels are plotted at the CCLM_ERA5_CEU-3 horizontal resolution. KYRILL is the strongest storm in our database.



# KYRILL Storm Wind Gust Differences

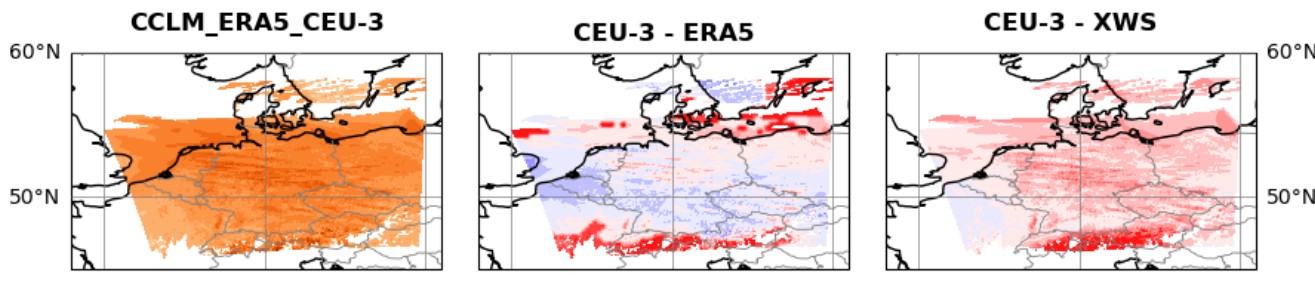

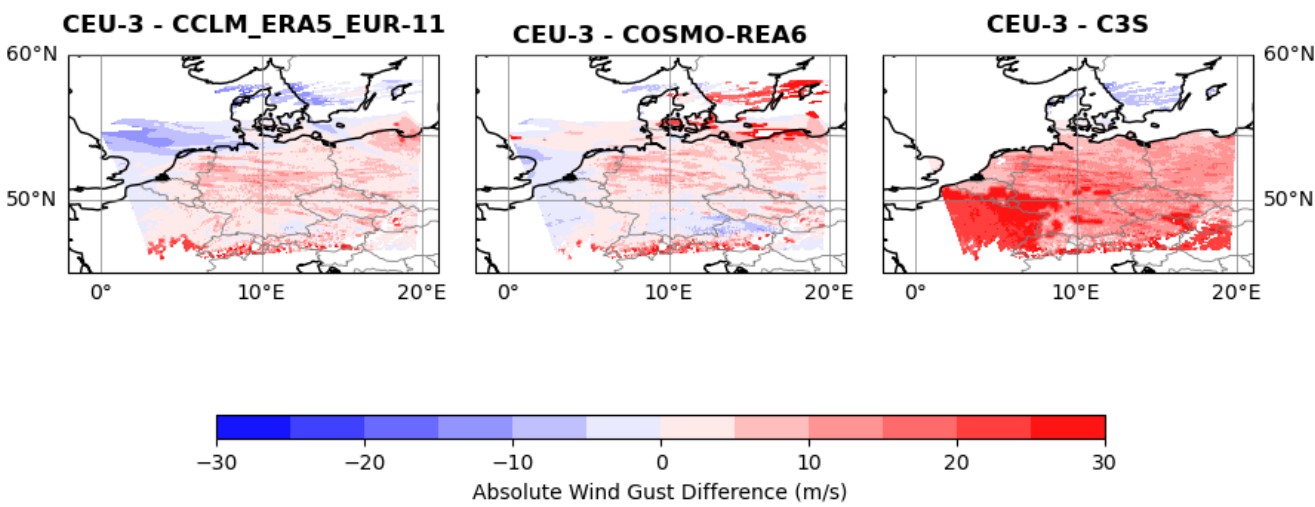

**Figure 10.** The daily maximum wind gusts (ms$^{-1}$) for windstorm KYRILL (Jan 2007) associated with its footprint as derived from CCLM_ERA5_CEU-3 and shown in Fig. 2, displayed here in the top left panel. The five remaining panels display the difference in ms$^{-1}$ between the CCLM_ERA5_CEU-3 absolute wind gusts and the wind gusts derived from ERA5, XWS, C3S, COSMO-REA6, and CCLM_ERA5_EUR-11 after regridding to the CCLM_ERA5_CEU-3 resolution, in clockwise order. Colors for the CCLM_ERA5_CEU-3 wind gusts are the same as in Figure 2, while the the colors representing the wind gust differences displayed in the remaining five panels are defined by the color bar below the plot. All panels are plotted at the CCLM_ERA5_CEU-3 horizontal resolution. KYRILL is the strongest storm in our database.

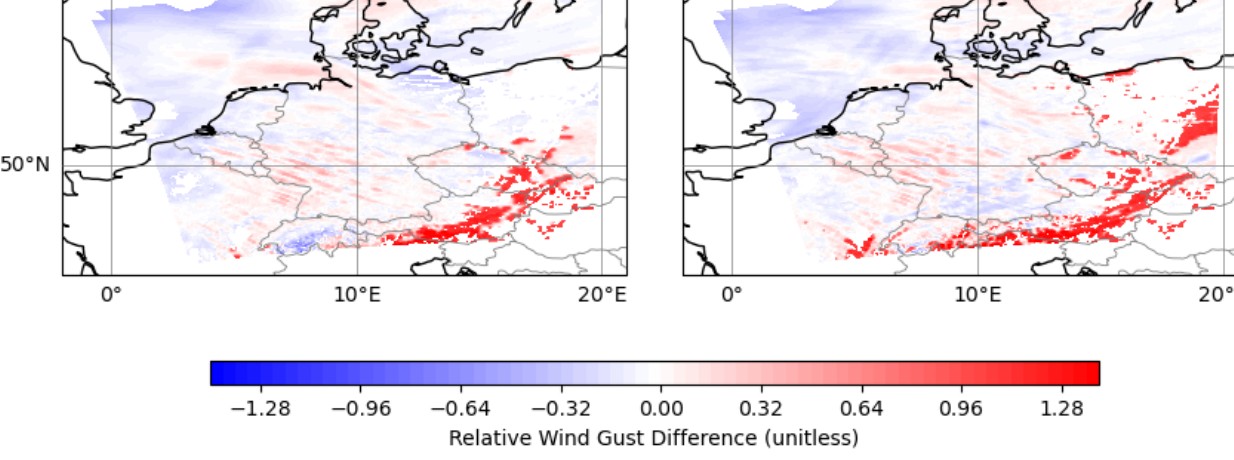

**Figure 11.** Storm footprint for windstorm ANDREA (Jan 2012) derived from the CCLM_ERA5_CEU-3 data set in the top left panel. The three remaining panels display the difference between the CCLM_ERA5_CEU-3 footprint and the footprints derived from ERA5, COSMO-REA6, and CCLM_ERA5_EUR-11 after regridding to the CCLM_ERA5_CEU-3 resolution, in clockwise order. Colors for the CCLM_ERA5_CEU-3 footprint are the same as in Figure 1, while the the colors representing the footprint differences displayed in the remaining three panels are defined by the color bar below the plot. All panels are plotted at the CCLM_ERA5_CEU-3 horizontal resolution. ANDREA is one of the stronger storms in our database.



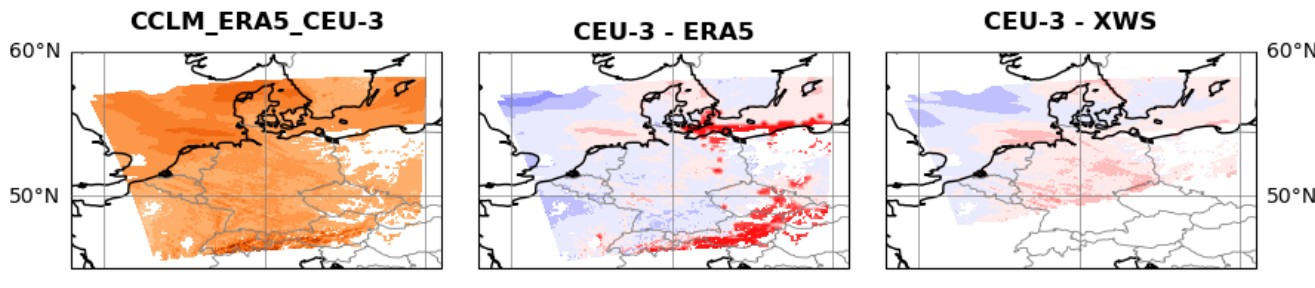

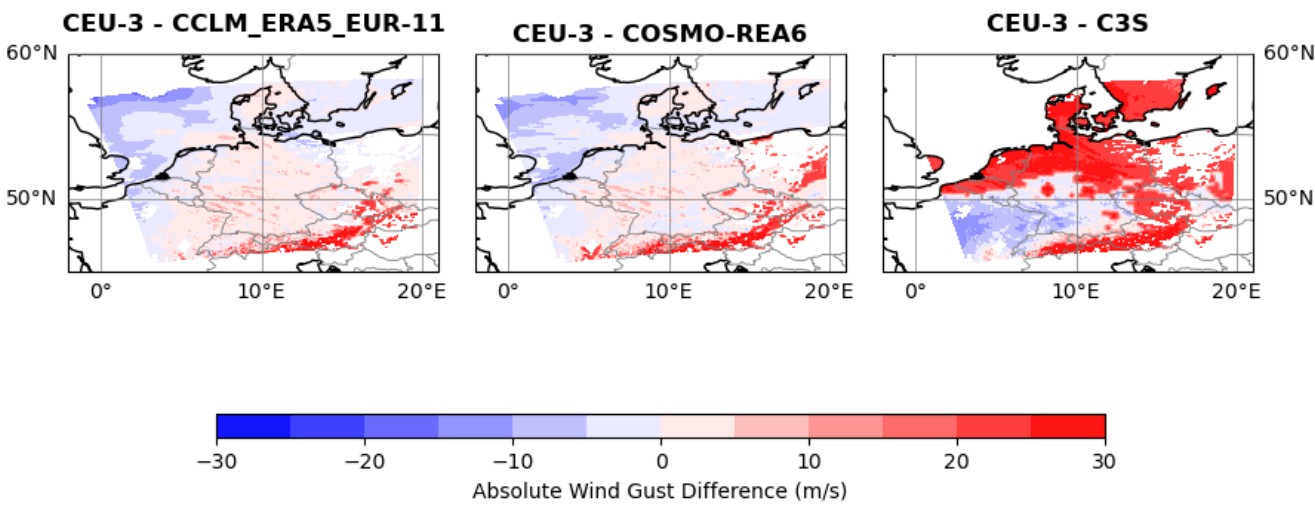

**Figure 12.** The daily maximum wind gusts (ms$^{-1}$) for windstorm ANDREA (Jan 2012) associated with its footprint as derived from CCLM_ERA5_CEU-3, displayed here in the top left panel. The five remaining panels display the difference in ms$^{-1}$ between the CCLM_ERA5_CEU-3 absolute wind gusts and the wind gusts derived from ERA5, XWS, C3S, COSMO-REA6, and CCLM_ERA5_EUR-11 after regridding to the CCLM_ERA5_CEU-3 resolution, in clockwise order. Colors for the CCLM_ERA5_CEU-3 wind gusts are the same as in Figure 2, while the the colors representing the wind gust differences displayed in the remaining five panels are defined by the color bar below the plot. All panels are plotted at the CCLM_ERA5_CEU-3 horizontal resolution. ANDREA is one of the stronger storms in our database.