# Peer review of "CLIMK-WINDS: A New Database of Extreme European Winter Windstorms"

_Earth System Science Data, 2024_

## Referee Comment (RC2)

**Review on "A new Database of Extreme European Winter Windstorms" by C.M. Flynn, J. Moemken, J. G. Pinto and G. Messori.**

Overall this is an interesting paper and well written. The study provides a useful database of wind footprints based on different meteorological input data that will likely be beneficial to both academic researchers and to stakeholders such as (re-)insurance companies. I do not have any major criticisms of this manuscript but I do have a number of minor comments which should be addressed. These are detailed below in order of line number rather than severity.

**Minor Comments**
1. Abstract. Currently this primarily states what has been done in this study and there is very little mention of the key results. I suggest that a sentence or two about the main differences in the wind footprints and identified top50 storms between the four different input datasets is included here.
2. Lines 16-17. "...allows to characterise the impact horizontal resolution can have on footprint identification and severity assessment". This is somewhat misleading as it is not just the horizontal resolution that differs between the 4 different input data sets. However, as CCLM_ERA5_CEU-3 and CCLM_ERA5_EUR-11 are the same model but run on different domains at different resolutions, this statement does hold if only those two input data sources are considered.
3. Lines 47-49. Please check the language / grammar here. Something does not seem correct.
4. Lines 99 – 103. At the start of section 2.1 the various diagnostics are introduced but not explained and I was left to wonder what they were. These are explained in section 2.3. I suggest adding something like "explained in more detail in section 2.3" to these lines.
5. Section 2.2. Could the domains covered by the four different input data sets be shown on a map? It is quite hard to mentally visualise these using just the latitude / longitude values especially when they are not presented together in the text.
6. Line 160-161. As the loss index depends on the exceedance of the $98^{th}$ percentile of the daily maximum wind gusts, I think it would be interesting and informative to include a figure showing how the $98^{th}$ percentile varies between the four different input data sets. If such a figure was included, this would also nicely show the spatial extent of the different domains hence addressing point number 5 above.
7. Line 171. As the loss index depends on the population density, could maps of this be also included in the manuscript?
8. Lines 234-235. How is the footprint size index, *N*, computed?
9. Line 242. Is the link here provided the correct one? It goes to the main climate data store homepage. Please check this. It may also be better to add this as a proper reference as is done on line 221.
10. Lines 252 – 256. I struggle to understand why the C3S wind footprints need to be masked. I think this is because I do not fully understand the explanation in lines 245 – 251 of how this dataset was produced. Can this please be clarified by adding some more details here.
11. Lines 284-285 and 296-297, "29 storms were identified within all four input sources". This is potentially misleading as it is very likely that some of the 47 storms that were identified in at least one but not all input data sets were actually present in the input data sets but they were not in the top 50 storms. I suggest the quoted text above is revised. However, it would also be very interesting to know where the 76 unique storms ranked in each of the 4 input datasets or if they were not detected at all (e.g. no wind gusts exceeding the $98^{th}$ percentile). This could be included by adding a table similar to table 2.
12. Line 302-303. Is this unexpected result due to compensating effects? e.g. CCLM_ERA5_CEU-3 has a smaller domain (potentially fewer storms) but higher resolution (more storms)?

13. Section 3.2 The comparison between the first 10- and second 10-years seems a bit arbitrary and the differences are likely not statistically robust. This analysis requires better motivation. Additionally, some more robust statistical analysis would strengthen this aspect of the manuscript. For example, temporal trends could be estimates or tipping points in the time series could be searched for.
14. Line 416 "the three databases". Is "three" a typo here? I'm not sure which databases are being referred to here.
15. Lines 425 – 428. First, it is not clear exactly how the mean of the footprint difference are computed. Is it that the mean of all 50 storms in each data set is computed and then the difference is taken between these means? Or is the difference between each data set done for each storm first, then the mean taken? Second, could the mean absolute error be used in addition as this would avoid the cancellation of errors problem as noted in line 429.
16. Section 3.4.2. The first paragraph of this section is more of a comparison between the spatial variability in the footprint and in the absolute wind gusts. This makes the heading of this subsection inaccurate. Possibly the authors want to re-consider the structure of this part of the manuscript.
17. Lines 459 and 460. It would be helpful to refer to specific figures / figure panels here.
18. Line 462. "in contrast to the mean footprint differences of small magnitude and variable sign". In Figure 7, top right panel, there is more red than blue so I disagree with the expression "variable sign" in this sentence.
19. Figure 2. Are the wind gusts from C3S plotted after the masking has been performed? Please add this information to the caption.
20. Figures 3, 4 and 5. These are missing y-labels. Additionally, they are rather noisy and hard to read. Would these be better as bar charts? Adding grid lines would also help.
21. Figure 5. Rather than having data gaps when there is no storm activity, could this somehow be indicated on the figure? e.g. extend the y-axis to lower values and have a y-tick mark stating "undefined" and then plot the data against that value on the y-axis?
22. Figure 6. This is very difficult to read. The pale-yellow colour for C3S is almost impossible to see. Could this figure be stretched in the y-direction to give more space for each bar?
23. Figure 8. Could the white space between the panels be reduced and the panel size increased?
24. Figures 9 -12. The top left panel in each of these figures is missing a colour bar. In Figure 9 and 10, these panels are repeats of panels from Figures 1 and 2, however, I still think a colour bar needs to be included in these figures.

---

## Author Response (AR1)

**Revision replies; see below for discussion replies.**

**Reviewer #1 (Hugo Rakotoarimanga)**

*The preprint is of good quality. The methodology is clearly exposed with appropriate references. The manuscript points towards applications to risk assessment and catastrophe modelling of a novel database of Extreme European windstorms. This dataset could specifically be useful to insurers with a European exposure, as the authors provide a clear understanding of the importance of reliable hazard data and the quantification of uncertainty which are current pain points in the practice of catastrophe modelling.*

*The dataset is novel as it aims at bridging a gap in the existing hazard sources concerning European winter windstorms by providing a consistent and robust approach in the modelling of extreme winds.*

*Contrasting with other available datasets such as XWS or C3S, one novelty of the dataset is the availability of several 'realizations' of each of the Top 50 European windstorms allowing for the quantification and exploration of uncertainties and their causes (such as horizontal resolution of the footprints).*

Dear Hugo, thank you for your positive feedback on our work and suggestions for improvement. We provide below detailed replies to your specific comments, outlining the edits that we have implemented in our revised submission.

1) *The proposed novel database should be given a name.*

This is a good point. In the revised version of the paper we refer to the database as the CLIMK–WINDS (CLimes IMK–WINDStorm) database. This reflects the content of the database and the two research institutions that have produced it: climes at Uppsala University and IMK at Karlsruhe Institute of Technology.

2) *The dataset covers the period 1995-2015 where most of the major storms have occurred in Europe, and the authors' claim is that the methodology could be easily applied to more recent storms. However, the data used to produce this new dataset seems to be available up to 2019. The reasons why the novel dataset only covers the period up to 2015 should be made explicit.*

You are correct in noting that most of the data that we use extends up to 2019. The reason for limiting our dataset to an earlier date was to enable a comparison with XWS, which is the only dataset in our analysis which makes use of insurance data.

We had thus initially decided to focus on the 1995-2014 20-year period, as that is the common period for all datasets. We then decided to add in 2015 because it contains 3 of our 29 common storms (>10% of the total). We deemed this an acceptable compromise between ensuring a fair comparison across datasets and covering a period of major storms affecting Europe. As you also note, the latter is a key goal of our database.

We agree that the above rationale was not explained in the text, and have revised Sect. 2.1 to clarify this. We have additionally updated Table 1 to include the period covered by each dataset. Finally, we have corrected the misleading visual formatting of Tables 2 and 3 and Figures 4 – 7 to avoid giving the impression that XWS covers 2015 but has zero Top50 storms in that year.

3) *The impact assessment of European windstorms is one of the main applications of the novel proposed dataset outlined by the manuscript. Hence the authors have chosen the well-established Loss Index to measure impacts, as there is no publicly available consistent economic or insured exposure database that could directly enable the computation of losses. However, I suggest that it may be more appropriate to explain that risk metrics such as OEP and AEP are proxy quantities, derived from the LI and not from actual economic or insured, for the sake of clarity. This comment is applicable to all mentions of losses.*

We fully agree that this is an important point to clarify. We now state explicitly at several points in the paper that the losses and the risk metrics that we quantify are all based on an empirical model which uses wind speed data rather than on actual loss data. We specifically added text to this effect at the end of the introduction , in Sect. 2.3, Sect. 2.6 and in the concluding section. Since we use the term "loss" or a variant of such a term over 100 times in the paper, we believe it would hinder readability if we were to change all of these references to refer to "proxy loss" or a similar wording. We have however added the term "proxy" in Sect. 2.4 when we describe the notion of aggregated loss (see also our reply to your Comment #10 below; lines 195-196).

4) *A general remark about the figures is that they are quite small, it is a bit difficult to assess visually the differences between the different data sources. For example, Figure 1 is quite clear, but the size of subplots in Figure 2 and the colorbar used for absolute wind gust makes visual appreciation of differences difficult. From an impact perspective it could be interesting to locate major cities (e.g. Cologne, Munich, Berlin…) on the maps.*

Thank you for these suggestions. We have increased the size of the subplots in Figures 2, 8, 10, and 12, and have changed the colormap of absolute wind gusts in Figures 3, 11, and 13 to better distinguish the wind gust speeds. Regarding your suggestion about locating major cities, we have decided not to show major cities on the maps of wind gusts, as the labels become very hard to read when superimposed on the background shading. However, we have included a new figure in the supplemental material showing the mean storm footprint Loss Index computed over the common storms, as well as maps of the population density for the grids of each input data set. The latter show clearly the locations of urban areas as high-density data points.

5) *I suggest that Figure 4 and 5 should be instead bar plots for readability and as the data represented here is discrete on the x-axis.*

You are correct in noting that the data is discrete along the time axis. We have replotted these figures as suggested.

6) *It would be interesting to the reader (especially with a focus on impacts) to represent the differences in terms of Loss Index between the different data sources on a figure, like the plots on Figure 2. This could be integrated at regional (e.g. administrative divisions) or country level.*

In line with your suggestion, we have added a figure showing the Loss Index for common storm footprints and the differences between the four data sets in the supplementary material. The differences are most pronounced over densely populated regions, which we partly ascribe to the fact that we regrid the population density data to match the horizontal resolution of the wind input file.

7) *Figure 6 is difficult to read. It is informative indeed to keep this kind of figure to compare the differences in terms of normalized loss among the different data sources. Maybe this figure should only represent a fewer number of storms for readability (10?). I would suggest ordering the storms by decreasing normalized loss instead.*

We agree with your impression. We have split the figure into two panels, one showing the top 10 storms and the other showing all remaining storms. We have further ordered storms by decreasing normalized loss with respect to ERA5 .

*8) While some storm names are obvious and well-known, some are quite obscure such as storm Fridtjof where a simple web search does not yield much information. It could be useful to always add the year of occurrence next to the storm name for the general reader.*

Thank you for this good suggestion. We have updated the labels in Fig. 6 accordingly and have ensured that figures referring to specific storms always include the storm date in the caption.

*9) I suggest that a figure should represent the geographical extent of the different domains considered for clarity and their overlap. This is a very minor comment.*

We fully agree with this. In line with the comments 5 and 6 by Reviewer #2, we have added a figure showing the 98th percentile of daily maximum wind gusts for each of the four input data sets (new Fig. 1). This figure also highlights the geographical extent of the different domains.

*10) Line 103, we suggest replacing "integrated loss" by "integrated proxy of the losses" (see Specific comments).*

We have moved this passage to Sect. 2.4 and added a sentence in correspondence to this point, specifying that the integrated loss should be regarded as a proxy loss.

*11) Line 416, there seems to be a grammar mistake, "point" should instead be "points".*

Thank you for having spotted this grammatical mistake; we have corrected it.

**Reviewer #2 (Anonymous)**

*Overall this is an interesting paper and well written. The study provides a useful database of wind footprints based on different meteorological input data that will likely be beneficial to both academic researchers and to stakeholders such as (re-)insurance companies. I do not have any major criticisms of this manuscript but I do have a number of minor comments which should be addressed. These are detailed below in order of line number rather than severity.*

We thank the Reviewer for their positive feedback on our work and suggestions for improvement. We provide below detailed replies to the specific comments, outlining the edits that we have implemented in our revised submission.

1) *Abstract. Currently this primarily states what has been done in this study and there is very little mention of the key results. I suggest that a sentence or two about the main differences in the wind footprints and identified top50 storms between the four different input datasets is included here.*

The Reviewer raises a very good point. We have added an overview of the results to the abstract, and have concomitantly shortened the pre-existing text to keep the abstract concise.

2) *Lines 16-17. "...allows to characterise the impact horizontal resolution can have on footprint identification and severity assessment". This is somewhat misleading as it is not just the horizontal resolution that differs between the 4 different input data sets. However, as CCLM_ERA5_CEU-3 and CCLM_ERA5_EUR-11 are the same model but run on different domains at different resolutions, this statement does hold if only those two input data sources are considered.*

We understand the potential misunderstanding that this may lead to in the abstract, when the reader has not yet been introduced to the details of the different datasets that we analyse. We have rephrased this in the abstract to read: "the choice of input data set – including the data's horizontal resolution –". We hope that this clarifies that horizontal resolution is one of the many dimensions involved in the choice of dataset. The Reviewer is correct in stating that the effect of horizontal resolution is only isolated when comparing the footprints from the CCLM_ERA5_CEU-3 and CCLM_ERA5_EUR-11 datasets. We prefer not to go into this level of detail in the abstract. However, we now state this explicitly in the discussion in Sect. 4.

*3) Lines 47-49. Please check the language / grammar here. Something does not seem correct.*

We have rephrased this passage to: "Insurance and reinsurance companies also simulate storm losses using catastrophe models. However, the loss estimates based on storm severity indices or on catastrophe models, both require accurate wind speed or gust data with high spatial and temporal coverage."

*4) Lines 99 – 103. At the start of section 2.1 the various diagnostics are introduced but not explained and I was left to wonder what they were. These are explained in section 2.3. I suggest adding something like "explained in more detail in section 2.3" to these lines.*

We agree, and have added references to the different data and method subsections in Sect. 2.1, including a pointer to Sect. 2.3. We have further moved the short discussion of the unintegrated and integrated loss indices to Sect. 2.3, as we understand that such a discussion at this point in the paper would be difficult to follow for a reader that is not already familiar with the loss index.

*5) Section 2.2. Could the domains covered by the four different input data sets be shown on a map? It is quite hard to mentally visualise these using just the latitude / longitude values especially when they are not presented together in the text.*

This is a good suggestion. In line with the following comment, and with comment #9 by Reviewer #1, we have added a figure showing the 98th percentile of daily maximum wind gusts (new Fig. 1), which also highlights the different spatial extent of the four input data sets.

*6) Line 160-161. As the loss index depends on the exceedance of the 98th percentile of the daily maximum wind gusts, I think it would be interesting and informative to include a figure showing how the 98th percentile varies between the four different input data sets. If such a figure was included, this would also nicely show the spatial extent of the different domains hence addressing point number 5 above.*

Following your previous comment, we have added the requested figure to the manuscript. The 98th percentile exhibits some regional differences between the four data sets in terms of amplitude, e.g., along the east coast of the UK or over Sweden.

Additionally, differences arise due to the different resolutions of the data sets in regions with sharp topography, e.g., in the Alps or other mountainous regions.

7) *Line 171. As the loss index depends on the population density, could maps of this be also included in the manuscript?*

We agree that adding maps of the population density can help readers understand how the loss index may vary spatially. We have include maps of the re-gridded population density for each of the input data sets in the supplementary material.

8) *Lines 234-235. How is the footprint size index, N, computed?*

We have added a more detailed description of N to the text. N is defined as the number of 25 km footprint grid points over European and Scandinavian land for which the maximum wind gust exceeds 25 m s$^{-1}$.

9) *Line 242. Is the link here provided the correct one? It goes to the main climate data store homepage. Please check this. It may also be better to add this as a proper reference as is done on line 221.*

We thank the Reviewer for having noticed this mistake on our part. We have updated the statement to provide the reference in the text and the full link in the Data Availability Section.

10) *Lines 252 – 256. I struggle to understand why the C3S wind footprints need to be masked. I think this is because I do not fully understand the explanation in lines 245 – 251 of how this dataset was produced. Can this please be clarified by adding some more details here.*

We have rephrased and expanded the explanation as suggested. In C3S, the strongest wind gusts associated with a storm were estimated over all land areas within the full C3S domain, without distinguishing between those grid points that are and are not impacted by a given storm. No cut-off criterion, such as the 1000 km radius centered on the storm track used in the XWS database, was applied. This leaves ambiguity over precisely where the footprint of a storm begins and ends. Because the C3S database is derived from ERA5, we used the footprints we identified from the ERA5 data set in the course of creating our database as a spatial mask to "cut out" the C3S wind gusts belonging to a given storm's footprint, after first

regridding the ERA5 footprints to the C3S horizontal grid. This explanation is now included in the text.

11) *Lines 284-285 and 296-297, "29 storms were identified within all four input sources". This is potentially misleading as it is very likely that some of the 47 storms that were identified in at least one but not all input data sets were actually present in the input data sets but they were not in the top 50 storms. I suggest the quoted text above is revised. However, it would also be very interesting to know where the 76 unique storms ranked in each of the 4 input datasets or if they were not detected at all (e.g. no wind gusts exceeding the 98th percentile). This could be included by adding a table similar to table 2.*

We agree that our formulation was misleading, and have rephrased the two passages and some later references to the number of detected storms to highlight that we are speaking only of Top50 storms. A table corresponding to Table 2, which includes all 76 unique Top50 storms, is provided in the Supplementary Material. We now reference this at the beginning of Sect. 3.1, specifying that it also includes the ordinal rank of the storms within the respective dataset.

12) *Line 302-303. Is this unexpected result due to compensating effects? e.g. CCLM_ERA5_CEU-3 has a smaller domain (potentially fewer storms) but higher resolution (more storms)?*

We thank the Reviewer for proposing this hypothesis, which we now formulate in Sect. 4 (lines 540-541).

13) *Section 3.2 The comparison between the first 10- and second 10-years seems a bit arbitrary and the differences are likely not statistically robust. This analysis requires better motivation. Additionally, some more robust statistical analysis would strengthen this aspect of the manuscript. For example, temporal trends could be estimates or tipping points in the time series could be searched for.*

We have removed the comparison between the two decades since we agree that it lacks robustness. We are hesitant to conduct a tipping point analysis, as tipping points have specific mathematical-statistical properties, for example irreversibility, which it is not possible to assess based on the information provided in our database. Following the Reviewer's suggestion, we have however tested whether any significant temporal trends emerge. We find that all data sets display negative trends

in the number of storms and the associated damage, but that none of these trends is significant. We now mention this in the paper. We copy the full results below.

**Proportion of Top50 storms per winter**

(data set, p-value of linear fit, slope of linear fit)

ERA5, p-value: 0.23, slope: -0.13

CCLM_ERA5_EUR-11, p-value: 0.29, slope: -0.13

COSMO-REA6, p-value: 0.57, slope: -0.06

CCLM_ERA5_CEU-3, p-value: 0.22, slope: -0.12

XWS, p-value: 0.28, slope: -0.17

C3S, p-value: 0.57, slope: -0.09

**Proportion of total losses per winter**

ERA5, p-value: 0.28, slope: -0.13

CCLM_ERA5_EUR-11, p-value: 0.27, slope: -0.14

COSMO-REA6, p-value: 0.45, slope: -0.08

CCLM_ERA5_CEU-3, p-value: 0.18, slope: -0.14

XWS, p-value: 0.52, slope: -0.12

C3S, p-value: 0.44, slope: -0.12

14) *Line 416 "the three databases". Is "three" a typo here? I'm not sure which databases are being referred to here.*

We intended to refer to our database, XWS and C3S, but we agree that the formulation was misleading. We have rephrased and now spell out the names of the three databases that we intended to refer to.

15) *Lines 425 – 428. First, it is not clear exactly how the mean of the footprint difference are computed. Is it that the mean of all 50 storms in each data set is computed and then the difference is taken between these means? Or is the difference between each data set done for each storm first, then the mean*

*taken? Second, could the mean absolute error be used in addition as this would avoid the cancellation of errors problem as noted in line 429.*

Our approach is to take the difference between the footprints of each storm first, and then average across storms. We have added an explanation of this to the text (lines 430-431).

We have also followed the Reviewer's suggestion of computing the mean absolute error. In general, amplitudes are higher using the mean absolute error compared to the mean difference. We find an increase of about 0.2 to 0.4 in many regions in the relative wind gust differences and an increase of about 5 m/s in the absolute wind gust differences. As hypothesized by the Reviewer, this points to partial cancellation of errors using the mean difference. Nonetheless, the mean absolute error generally shows a strong agreement with the mean difference in terms of spatial patterns for most datasets. Some regional differences however emerge, for example for XWS and CCLM_ERA5_EUR-11 over the North Sea.

We have decided to show both plots, and have added the mean absolute error figure to the supplementary material. We further refer to it and discuss it in the main text.

16) *Section 3.4.2. The first paragraph of this section is more of a comparison between the spatial variability in the footprint and in the absolute wind gusts. This makes the heading of this subsection inaccurate. Possibly the authors want to re-consider the structure of this part of the manuscript.*

We have restructured subsections 3.4.1 and 3.4.2 to more clearly separate the analysis of differences in the wind footprints and in the wind gusts. While we still refer to the role of footprint differences in explaining differences in the wind gusts in Sect. 3.4.2, the results in this subsection are now focussed on the wind gusts, in keeping with the subsection's heading.

17) *Lines 459 and 460. It would be helpful to refer to specific figures / figure panels here.*

We have restructured this part of the text and added references to the figures (now Figs. 8 and 9) as suggested.

18) *Line 462. "in contrast to the mean footprint differences of small magnitude and variable sign". In Figure 7, top right panel, there is more red than blue so I disagree with the expression "variable sign" in this sentence.*

We agree and have removed this formulation from the text.

19) *Figure 2. Are the wind gusts from C3S plotted after the masking has been performed? Please add this information to the caption.*

We have indeed plotted the wind gusts after performing the masking, and have added this information to the figure caption.

20) *Figures 3, 4 and 5. These are missing y-labels. Additionally, they are rather noisy and hard to read. Would these be better as bar charts? Adding grid lines would also help.*

We have added both x- and y-labels to all three figures. We have additionally redesigned the figures in the form of bar charts, also in response to comment 5 from Reviewer #1.

21) *Figure 5. Rather than having data gaps when there is no storm activity, could this somehow be indicated on the figure? e.g. extend the y-axis to lower values and have a y-tick mark stating "undefined" and then plot the data against that value on the y-axis?*

In response to the previous comment, we have redesigned the figure. We show no bars for years without storm activity, and explicitly state in the figure caption that there are years without storms in the database, which are reflected by no bar being shown for the respective data set.

22) *Figure 6. This is very difficult to read. The pale-yellow colour for C3S is almost impossible to see. Could this figure be stretched in the y-direction to give more space for each bar?*

We agree with your impression. We have split the figure into two panels, one showing the top 10 storms and the other showing all remaining storms. We have further ordered storms by decreasing normalized loss with respect to ERA5 .

23) *Figure 8. Could the white space between the panels be reduced and the panel size increased?*

We have increased the size of subplots in Figures 2, 8, 10, and 12.

> *24) Figures 9 -12. The top left panel in each of these figures is missing a colour bar. In Figure 9 and 10, these panels are repeats of panels from Figures 1 and 2, however, I still think a colour bar needs to be included in these figures.*

Thank you for spotting the missing colorbars in Figures 9-12. We have added these in.

**Discussion replies**

**Reviewer #1 (Hugo Rakotoarimanga)**

*The preprint is of good quality. The methodology is clearly exposed with appropriate references. The manuscript points towards applications to risk assessment and catastrophe modelling of a novel database of Extreme European windstorms. This dataset could specifically be useful to insurers with a European exposure, as the authors provide a clear understanding of the importance of reliable hazard data and the quantification of uncertainty which are current pain points in the practice of catastrophe modelling.*

*The dataset is novel as it aims at bridging a gap in the existing hazard sources*

*concerning European winter windstorms by providing a consistent and robust approach in the modelling of extreme winds.*

*Contrasting with other available datasets such as XWS or C3S, one novelty of the dataset is the availability of several 'realizations' of each of the Top 50 European windstorms allowing for the quantification and exploration of uncertainties and their causes (such as horizontal resolution of the footprints).*

Dear Hugo, thank you for your positive feedback on our work and suggestions for improvement. We provide below detailed replies to your specific comments, outlining the edits that we will implement in our revised submission.

1) *The proposed novel database should be given a name.*

This is a good point. In the revised version of the paper we will refer to the database as CLIMK–WINDS (CLimes IMK–WINDStorm) database. This reflects the content of the database and the two research institutions that have produced it: climes at Uppsala University and IMK at Karlsruhe Institute of Technology.

2) *The dataset covers the period 1995-2015 where most of the major storms have occurred in Europe, and the authors' claim is that the methodology could be easily applied to more recent storms. However, the data used to produce this new dataset seems to be available up to 2019. The reasons why the novel dataset only covers the period up to 2015 should be made explicit.*

You are correct in noting that most of the data that we use extends up to 2019. The reason for limiting our dataset to an earlier date was to enable a comparison with XWS, which is the only dataset in our analysis which makes use of insurance data. We had thus initially decided to focus on the 1995-2014 20-year period, as that is the common period for all datasets. We then decided to add in 2015 because it contains 3 of our 29 common storms (>10% of the total). We deemed this an acceptable compromise between ensuring a fair comparison across datasets and covering a period of major storms affecting Europe. As you also note, the latter is a key goal of our database.

We agree that the above rationale was not explained in the text, and will revise Sect. 2.1 to clarify this. We will additionally update Table 1 to include the period covered by each dataset. Finally, we will correct the misleading visual formatting of Table 2 and Figures 3 and 4 to avoid giving the impression that XWS covers 2015 but has zero Top50 storms in that year.

*3) The impact assessment of European windstorms is one of the main applications of the novel proposed dataset outlined by the manuscript. Hence the authors have chosen the well-established Loss Index to measure impacts, as there is no publicly available consistent economic or insured exposure database that could directly enable the computation of losses. However, I suggest that it may be more appropriate to explain that risk metrics such as OEP and AEP are proxy quantities, derived from the LI and not from actual economic or insured, for the sake of clarity. This comment is applicable to all mentions of losses.*

We fully agree that this is an important point to clarify. We will state explicitly at several points in the paper that the losses and the risk metrics that we quantify are all based on an empirical model which uses wind speed data rather than on actual loss data. We will specifically add text to this effect at the end of the introduction , in Sect. 2.3, Sect. 2.6 and in the concluding section. Since we use the term "loss" or a variant of such a term over 100 times in the paper, we believe it would hinder readability if we were to change all of these references to refer to "proxy loss" or a similar wording. We will however add the term "proxy" in Sect. 2.4 when we describe the notion of aggregated loss (see also our reply to your Comment #10 below).

*4) A general remark about the figures is that they are quite small, it is a bit difficult to assess visually the differences between the different data sources. For example, Figure 1 is quite clear, but the size of subplots in Figure 2 and the colorbar used for absolute wind gust makes visual appreciation of differences difficult. From an impact perspective it could be interesting to locate major cities (e.g. Cologne, Munich, Berlin…) on the maps.*

Thank you for these suggestions. We will increase the size of the subplots in Figures 2, 8, 10, and 12, and will change the colormap of absolute wind gusts in Figures 2, 10, and 12 to better distinguish the wind gust speeds. Regarding your suggestion about locating major cities, we have decided not to show major cities on the maps of wind gusts, as the labels become very hard to read when superimposed on the background shading. However, we will include a new figure in the supplemental material showing the mean storm footprint Loss Index computed over the common storms, as well as maps of the population density for each respective input data set.

*5) I suggest that Figure 4 and 5 should be instead bar plots for readability and as the data represented here is discrete on the x-axis.*

You are correct in noting that the data is discrete along the time axis. We will redo these figures as suggested.

*6) It would be interesting to the reader (especially with a focus on impacts) to represent the differences in terms of Loss Index between the different data sources on a figure, like the plots on Figure 2. This could be integrated at regional (e.g. administrative divisions) or country level.*

In line with your suggestion, we will add a figure showing the Loss Index for common storm footprints and the differences between the four data sets in the supplementary material. The differences are most pronounced over densely populated regions, which we partly ascribe to the fact that we regrid the population density data to match the horizontal resolution of the wind input file.

*7) Figure 6 is difficult to read. It is informative indeed to keep this kind of figure to compare the differences in terms of normalized loss among the different data sources. Maybe this figure should only represent a fewer number of storms for readability (10?). I would suggest ordering the storms by decreasing normalized loss instead.*

We agree with your impression. We will split the figure into two panels, one showing the top 10 storms and the other showing all remaining storms. We have further ordered storms by decreasing normalized loss with respect to ERA5 .

*8) While some storm names are obvious and well-known, some are quite obscure such as storm Fridtjof where a simple web search does not yield much information. It could be useful to always add the year of occurrence next to the storm name for the general reader.*

Thank you for this good suggestion. We will update the labels in Fig. 6 accordingly and ensure that figures referring to specific storms always include the storm date in the caption.

*9) I suggest that a figure should represent the geographical extent of the different domains considered for clarity and their overlap. This is a very minor comment.*

We fully agree with this. In line with the comments 5 and 6 by Reviewer #2, we will add a figure showing the 98th percentile of daily maximum wind gusts for each of the four input data sets. This figure will also highlight the geographical extent of the different domains.

*10) Line 103, we suggest replacing "integrated loss" by "integrated proxy of the losses" (see Specific comments).*

We will move this passage to Sect. 2.4 and add a sentence in correspondence to this point, specifying that the integrated loss should be regarded as a proxy loss.

*11) Line 416, there seems to be a grammar mistake, "point" should instead be "points".*

Thank you for having spotted this grammatical mistake; we will correct it.

**Reviewer #2 (Anonymous)**

*Overall this is an interesting paper and well written. The study provides a useful database of wind footprints based on different meteorological input data that will likely be beneficial to both academic researchers and to stakeholders such as (re-)insurance companies. I do not have any major criticisms of this manuscript but I do have a number of minor comments which should be addressed. These are detailed below in order of line number rather than severity.*

We thank the Reviewer for their positive feedback on our work and suggestions for improvement. We provide below detailed replies to the specific comments, outlining the edits that we will implement in our revised submission.

*1) Abstract. Currently this primarily states what has been done in this study and there is very little mention of the key results. I suggest that a sentence or two*

*about the main differences in the wind footprints and identified top50 storms between the four different input datasets is included here.*

The Reviewer raises a very good point. We will add an overview of the results to the abstract, and have concomitantly shortened the pre-existing text to keep the abstract concise.

2) *Lines 16-17. "...allows to characterise the impact horizontal resolution can have on footprint identification and severity assessment". This is somewhat misleading as it is not just the horizontal resolution that differs between the 4 different input data sets. However, as CCLM_ERA5_CEU-3 and CCLM_ERA5_EUR-11 are the same model but run on different domains at different resolutions, this statement does hold if only those two input data sources are considered.*

We understand the potential misunderstanding that this may lead to in the abstract, when the reader has not yet been introduced to the details of the different datasets that we analyse. We will rephrase this in the abstract to read: "the choice of input data set – including the data's horizontal resolution –". We hope that this clarifies that horizontal resolution is one of the many dimensions involved in the choice of dataset. The Reviewer is correct in stating that the effect of horizontal resolution is only isolated when comparing the footprints from the CCLM_ERA5_CEU-3 and CCLM_ERA5_EUR-11 datasets. We prefer not to go into this level of detail in the abstract. However, we will state this explicitly in the discussion in Sect. 4.

3) *Lines 47-49. Please check the language / grammar here. Something does not seem correct.*

We will rephrase this passage to: "Insurance and reinsurance companies also simulate storm losses using catastrophe models. However, the loss estimates based on storm severity indices or on catastrophe modelling, both require accurate wind speed or gust data with high spatial and temporal coverage."

4) *Lines 99 – 103. At the start of section 2.1 the various diagnostics are introduced but not explained and I was left to wonder what they were. These are explained in section 2.3. I suggest adding something like "explained in more detail in section 2.3" to these lines.*

We agree, and will add references to the different data and method subsections in Sect. 2.1, including a pointer to Sect. 2.3. We will further move the short discussion

of the unintegrated and integrated loss indices to Sect. 2.3, as we understand that such a discussion at this point in the paper would be difficult to follow for a reader that is not already familiar with the loss index.

5) *Section 2.2. Could the domains covered by the four different input data sets be shown on a map? It is quite hard to mentally visualise these using just the latitude / longitude values especially when they are not presented together in the text.*

This is a good suggestion. In line with the following comment, and with comment #9 by Reviewer #1, we will add a figure showing the 98th percentile of daily maximum wind gusts, which also highlights the different spatial extent of the four input data sets.

6) *Line 160-161. As the loss index depends on the exceedance of the 98th percentile of the daily maximum wind gusts, I think it would be interesting and informative to include a figure showing how the 98th percentile varies between the four different input data sets. If such a figure was included, this would also nicely show the spatial extent of the different domains hence addressing point number 5 above.*

Following from your previous comment, we will add the requested figure to the manuscript. The 98th percentile exhibits some regional differences between the four data sets in terms of amplitude, e.g., at the east coast of the UK or Sweden. Additionally, differences arise due to the different resolutions of the data sets in regions with sharp topography, e.g., in the Alps or other mountainous regions.

7) *Line 171. As the loss index depends on the population density, could maps of this be also included in the manuscript?*

We agree that adding maps of the population density can help readers understand how the loss index may vary spatially. We will include maps of the re-gridded population density for each of the input data sets in the supplementary material.

8) *Lines 234-235. How is the footprint size index, N, computed?*

We will add a more detailed description of N to the text. N is defined as the number of 25 km footprint grid points over European and Scandinavian land for which the maximum wind gust exceeds 25 m s$^{-1}$.

9) *Line 242. Is the link here provided the correct one? It goes to the main climate data store homepage. Please check this. It may also be better to add this as a proper reference as is done on line 221.*

We thank the Reviewer for having noticed this mistake on our part. We will update the statement to provide the reference in the text and the full link in the Data Availability Section.

10) *Lines 252 – 256. I struggle to understand why the C3S wind footprints need to be masked. I think this is because I do not fully understand the explanation in lines 245 – 251 of how this dataset was produced. Can this please be clarified by adding some more details here.*

We will rephrase and expand the explanation as suggested. In C3S, the strongest wind gusts associated with a storm were estimated over all land areas within the full C3S domain, without distinguishing between those grid points that are and are not impacted by a given storm. No cut-off criterion, such as the 1000 km radius centered on the storm track used in the XWS database, was applied. This leaves ambiguity over precisely where the footprint of a storm begins and ends. Because the C3S database is derived from ERA5, we used the footprints we identified from the ERA5 data set in the course of creating our database as a spatial mask to "cut out" the C3S wind gusts belonging to a given storm's footprint, after first regridding the ERA5 footprints to the C3S horizontal grid. This explanation will be included in the text.

11) *Lines 284-285 and 296-297, "29 storms were identified within all four input sources". This is potentially misleading as it is very likely that some of the 47 storms that were identified in at least one but not all input data sets were actually present in the input data sets but they were not in the top 50 storms. I suggest the quoted text above is revised. However, it would also be very interesting to know where the 76 unique storms ranked in each of the 4 input datasets or if they were not detected at all (e.g. no wind gusts exceeding the 98th percentile). This could be included by adding a table similar to table 2.*

We agree that our formulation was misleading, and will rephrase the two passages and some later references to the number of detected storms to highlight that we are speaking only of Top50 storms. A table corresponding to Table 2, which includes all 76 unique Top50 storms, is provided in the Supplementary Material. We will reference this at the beginning of Sect. 3.1, specifying that it also includes the ordinal rank of the storms within the respective dataset.

12) *Line 302-303. Is this unexpected result due to compensating effects? e.g. CCLM_ERA5_CEU-3 has a smaller domain (potentially fewer storms) but higher resolution (more storms)?*

We thank the Reviewer for proposing this hypothesis, which we will formulate in Sect. 4.

13) *Section 3.2 The comparison between the first 10- and second 10-years seems a bit arbitrary and the differences are likely not statistically robust. This analysis requires better motivation. Additionally, some more robust statistical analysis would strengthen this aspect of the manuscript. For example, temporal trends could be estimates or tipping points in the time series could be searched for.*

We will remove the comparison between the two decades since we agree that it lacks robustness. Following the Reviewer's suggestion, we will verify whether any significant temporal trends emerge. We are hesitant to conduct a tipping point analysis, as tipping points have specific mathematical-statistical properties, for example irreversibility, which it is not possible to assess based on the information provided in our database.

14) *Line 416 "the three databases". Is "three" a typo here? I'm not sure which databases are being referred to here.*

We intended to refer to our database, XWS and C3S, but we agree that the formulation was misleading. We will rephrase and spell out the names of the three databases that we intended to refer to.

15) *Lines 425 – 428. First, it is not clear exactly how the mean of the footprint difference are computed. Is it that the mean of all 50 storms in each data set is computed and then the difference is taken between these means? Or is the difference between each data set done for each storm first, then the mean taken? Second, could the mean absolute error be used in addition as this would avoid the cancellation of errors problem as noted in line 429.*

Our approach is to take the difference between the footprints of each storm first, and then average across storms. We will add an explanation of this to the text.

We have also followed the Reviewer's suggestion of computing the mean absolute error. In general, amplitudes are higher using the mean absolute error compared to the mean difference. We find an increase of about 0.2 to 0.4 in many regions in the relative wind gust differences and an increase of about 5 m/s in the absolute wind gust differences. As hypothesized by the Reviewer, this points to partial cancellation of errors using the mean difference. Nonetheless, the mean absolute error generally shows a strong agreement with the mean difference in terms of spatial patterns for most datasets. Some regional differences emerge, for example for XWS and CCLM_ERA5_EUR-11 over the North Sea.

We will evaluate whether to substitute the mean difference plots with mean absolute error in the paper, or whether to show both.

16) *Section 3.4.2. The first paragraph of this section is more of a comparison between the spatial variability in the footprint and in the absolute wind gusts. This makes the heading of this subsection inaccurate. Possibly the authors want to re-consider the structure of this part of the manuscript.*

We find it logical to discuss differences in wind gusts in the context of differences in the windstorm footprints. At the same time, we agree that the context of the subsection does not reflect the heading. We will therefore revise the subsection heading to: "". We will further restructure the subsection to more clearly separate the analysis of differences in the wind gusts and of the role of footprint differences in explaining differences in the wind gusts.

17) *Lines 459 and 460. It would be helpful to refer to specific figures / figure panels here.*

We will add references to the top left panels in Figs. 7 and 8, respectively.

18) *Line 462. "in contrast to the mean footprint differences of small magnitude and variable sign". In Figure 7, top right panel, there is more red than blue so I disagree with the expression "variable sign" in this sentence.*

We agree and will rephrase this to: "weakly positive or near-zero".

19) *Figure 2. Are the wind gusts from C3S plotted after the masking has been performed? Please add this information to the caption.*

We have indeed plotted the wind gusts after performing the masking, and will add this information to the figure caption.

20) *Figures 3, 4 and 5. These are missing y-labels. Additionally, they are rather noisy and hard to read. Would these be better as bar charts? Adding grid lines would also help.*

We will add y-lables reading "Winter Season" to all three figures. We will additionally redesign the figures in the form of bar charts, also in response to comment 5 from Reviewer #1.

21) *Figure 5. Rather than having data gaps when there is no storm activity, could this somehow be indicated on the figure? e.g. extend the y-axis to lower values and have a y-tick mark stating "undefined" and then plot the data against that value on the y-axis?*

In response to the previous comment, we will redesign the figure. We will show no bars for years without storm activity, and will also explicitly state in the figure caption that there are years without storms in the database, which are reflected by no bar being shown for the respective data set.

22) *Figure 6. This is very difficult to read. The pale-yellow colour for C3S is almost impossible to see. Could this figure be stretched in the y-direction to give more space for each bar?*

We agree with your impression. We will split the figure into two panels, one showing the top 10 storms and the other showing all remaining storms. We will further order storms by decreasing normalized loss with respect to ERA5 .

23) *Figure 8. Could the white space between the panels be reduced and the panel size increased?*

We will increase the size of subplots in Figures 2, 8, 10, and 12.

*24) Figures 9 -12. The top left panel in each of these figures is missing a colour bar. In Figure 9 and 10, these panels are repeats of panels from Figures 1 and 2, however, I still think a colour bar needs to be included in these figures.*

Thank you for spotting the missing colorbar in Figures 9-12. We will add this in.